# Additive Manufacturing of Biomaterials—Design Principles and Their Implementation

**DOI:** 10.3390/ma15155457

**Published:** 2022-08-08

**Authors:** Mohammad J. Mirzaali, Vahid Moosabeiki, Seyed Mohammad Rajaai, Jie Zhou, Amir A. Zadpoor

**Affiliations:** Department of Biomechanical Engineering, Faculty of Mechanical, Maritime, and Materials Engineering, Delft University of Technology (TU Delft), Mekelweg 2, 2628 CD Delft, The Netherlands

**Keywords:** additive manufacturing, biomaterials, metals, polymers, ceramics

## Abstract

Additive manufacturing (AM, also known as 3D printing) is an advanced manufacturing technique that has enabled progress in the design and fabrication of customised or patient-specific (meta-)biomaterials and biomedical devices (e.g., implants, prosthetics, and orthotics) with complex internal microstructures and tuneable properties. In the past few decades, several design guidelines have been proposed for creating porous lattice structures, particularly for biomedical applications. Meanwhile, the capabilities of AM to fabricate a wide range of biomaterials, including metals and their alloys, polymers, and ceramics, have been exploited, offering unprecedented benefits to medical professionals and patients alike. In this review article, we provide an overview of the design principles that have been developed and used for the AM of biomaterials as well as those dealing with three major categories of biomaterials, i.e., metals (and their alloys), polymers, and ceramics. The design strategies can be categorised as: library-based design, topology optimisation, bio-inspired design, and meta-biomaterials. Recent developments related to the biomedical applications and fabrication methods of AM aimed at enhancing the quality of final 3D-printed biomaterials and improving their physical, mechanical, and biological characteristics are also highlighted. Finally, examples of 3D-printed biomaterials with tuned properties and functionalities are presented.

## 1. Introduction

Additive manufacturing (AM, also known as 3D printing) technologies are among the most feasible advanced manufacturing options to create complex structures for use in technology-driven industries, such as healthcare [1], automotive [2,3], and aerospace [4]. AM, being different from other manufacturing methods, such as subtractive and formative methods, results in less scrap and waste of materials and allows for lightweight complex structures, often hollow or porous, thus requiring less material input and energy input during their fabrication and service. Seven categories of AM, namely, binder jetting, directed energy deposition, material extrusion, material jetting, powder bed fusion, sheet lamination, and vat photopolymerisation, have been recognised and defined in the ISO/ASTM 52900 standard [5].

Not all AM processes in the ASTM classification are equally developed and used for medical devices and biomaterial fabrication [6]. Here, we summarise the capabilities, limitations, and pros and cons of conventional processes and associated materials (e.g., metals and their alloys, polymers, and ceramics) used in the fabrication of biomaterials (Table 1) in terms of printing speed, part sizes, degree of anisotropy, achievable resolution, the possibility of embedding cells in feedstock materials, the need for support, the need for post-processing, and costs. The success of each of these 3D printing processes relies, to a large extent, on the employment of optimised or suitable process parameters within the capabilities of the available AM machines that are associated with specific AM processes.

In addition to selecting the proper AM techniques and suitable printing parameters, the microarchitecture design of biomaterials is one of the critical aspects of their development. It is often necessary to design porous or lattice structures for biomedical applications. This implies that the morphologies and sizes of the pores of biomaterials must be fully open and interconnected to allow for the transport of nutrients and oxygen to cells [6,7,8].

The advent of AM technologies has provided unique opportunities for the accurate arrangement of the sizes and internal architectures of pores at a microscopic level and to produce organic geometries with complex internal architectures and passages [9,10,11]. This is one of the most important merits of AM over conventional fabrication technologies, such as casting and moulding [12], in which the designer has virtually no control over the precise details of the internal geometries of porous materials. The main objective of this review article is to present a clear picture of how this technology can be applied for producing biomaterials with novel designs, what the challenges and limitations are, and where the technology is heading. We summarise the current design principles employed in the fabrication of AM biomaterials. We also review the applications of different AM processes in the fabrication of metallic, polymeric, and ceramic biomaterials. It is intended to stimulate the further development and widespread application of the technology to turn design ideas into implants and other medical devices, as well as those of tissue engineering applications.

## 2. Geometrical Design of Lattices

While AM offers almost unlimited possibilities to part designers, there are several constraints in the structural design of lattices that limit the theoretical ability of AM to fabricate porous structures with highly complex geometries. Several inherent limitations related to the processability of the designed part also exist in AM methods, which has led to the introduction of several guidelines to manage these constraints and limitations [13]. Some of these constraints are recognised as minimum feature size (e.g., wall thickness, edges, and corners), the orientation of lattice structures on the build plate for self-overhanging, support materials, and support removal [14].

As an example, in powder bed fusion (PBF) techniques, overhanging structures, which are defined as parts of lattice structures that are not self-supported, can result in undesirable defects in lattice structures [15,16]. There are no underlying layers or solidified sections to support these overhanging parts during their fabrication, which is why the choice of orientation during building is critically important. The overhanging structure also depends on the critical fabrication angle [15]. Sacrificial support materials, therefore, need to be used for overhanging structures below a specific fabrication angle. These sacrificial support materials need to be removed (e.g., in PBF techniques) or washed away (e.g., in vat photopolymerisation techniques) from the structures during post-processing, which may damage additively manufactured parts. To compensate for that and achieve optimum results with fewer support materials, the parts need to be designed with self-supported struts in lattice structures. Restricted build envelopes and the application of a single material in the manufacturing process of metallic materials can also be specified as other limitations, although achievable sizes have been considerably increased in recent years, and combinations of materials have become possible, e.g., by means of a recoater. In some cases, the limitation of a combination of materials can be resolved by alloying elemental metallic powders [17]. This limitation can also be overcome by using multiple nozzles in extrusion-based AM techniques.

Creating the geometrical design of a lattice structure is the first step in designing AM lattices. Lattice structures can be broadly classified as open-cell or closed-cell cellular structures. Because it is not possible to remove the residual material (e.g., entrapped powder particles in the case of PBF processes or supports in vat photopolymerisation processes) in closed-cell lattices, open-cell lattices are mostly chosen for fabrication using AM techniques. There are various proposed design principles regarding the geometrical arrangement of lattice structures (an overview is provided in Table 2), which are discussed in detail in Section 2. In some cases, we may combine two or more of these design methods to obtain a more desirable lattice structure.

### 2.1. Library-Based Design

Computer-Aided Design (CAD), implicit surfaces, and image-based design can be categorised as traditional design strategies [18]. Open-source or commercial CAD tools/software have been used to develop CAD-based designs. These designs may then be transformed into the standard tessellation language (STL) format before going through the manufacturing process. In some cases, STL files can also be accessed through a software package installed on the 3D printing machine in order to control or modify the process parameters prior to or during printing. The final AM lattice structures can be generated by adjusting the process parameters of the input design file and setting the support material within the entire porous media.

Recently, other approaches (e.g., the single point exposure scanning strategy [19] and vector-based approach [20] for selective laser melting (SLM) printing or voxel-based approach [21] for Polyjet printing) have been proposed, which can boost the fabrication speed of an object with even more geometrical complexities. This is because the STL files of designs with too many complexities and details are often very large. The designs resulting from these approaches usually have smaller file sizes, thus allowing for easier file manipulation. These approaches, therefore, enable the process engineer to load large files with detailed features in the 3D printing software.

A unit cell can be identified as the smallest feature size in lattice structures with periodic microstructures. Unit cells create an ordered design by tessellating in a 2.5D plane (i.e., extruded in a 2D plane) or 3D space. Unit cells have already been identified in various forms, such as cubic or prismatic unit cells. They can be broadly categorised into two major groups, namely, beam-based and sheet-based unit cells. No specific repeating unit cells can be seen in lattices with irregular or random microstructures.

#### 2.1.1. Beam-Based Unit Cells

One of the most common geometries for producing metallic or non-metallic lattice structures is the beam- or strut-based design (Figure 1a), which includes beam-based unit cells that repeat spatially in 3D space. By reshaping the geometry, for example, by changing the size and thickness of struts and reforming the topology or connectivity of recurrent unit cells, the overall physical characteristics of the lattices, such as the relative density, pore size, and pore geometry, can be adjusted accordingly [22,23]. Body-centred cubic (BCC), face-centred cubic (FCC), and their variations (analogous to crystalline structures [24,25], cubic, diamond, and octet-truss) are just some examples of well-known strut-based topologies [26].

From a micro-mechanical viewpoint, lattice structures can be classified into two categories, namely, bending-dominated and stretching-dominated unit cells. Stretching-dominated unit cells are typically stiffer and have higher mechanical strength than bending-dominated ones [27]. However, achieving a fully stretch-dominated unit cell is nearly impossible, as some areas of the struts in a unit cell can experience bending loads. Strut-based unit cells can be characterised by their Maxwell number [28].

#### 2.1.2. Surface-Based Unit Cells

Sheet-based unit cells (Figure 1b) belong to the category of implicit surface designs, in which mathematical equations define pore configurations. Triply periodic minimal surfaces (TPMS) are specific classes of sheet-based unit cells that provide high flexibility in the design of lattice structures [38]. The full integration of pores in TPMS makes them suitable for use in scaffold designs in tissue regeneration and tissue ingrowth applications [38,39,40]. TPMS-based porous structures also have a zero-mean surface curvature that can be considered a unique property [8]. It must be emphasised that the fabrication of additively manufactured TPMS geometries with high quality is a challenging procedure. This limits the number of available TPMS designs with limited porosity. Some TPMS geometries, such as primitive, I-WP, gyroid, and diamond designs, can nevertheless be realised.

#### 2.1.3. Disordered and Random Network Designs

The arrangement of unit cells in lattice structures can be disordered, where the types or dimensions of the cells change within the object (Figure 1c). As an example of such disordered systems, functionally graded structures can be designed, where pore sizes vary within the lattices. AM of graded porous structures has recently become prevalent [41,42], particularly in biomedical engineering [43,44]. One crucial reason for this increasing interest is the feature that causes a smooth stress distribution in the product to avoid stress concentrations at abrupt geometrical alterations. However, their geometrical complexities cause the AM of graded arrangements to be challenging, particularly when they feature more stochastic or disordered graded designs. This can result in the manufacturing of struts that are incapable of self-support, resulting in a poor AM outcome.

In contrast to uniform lattice structures, disordered lattice structures have several advantages. First, they can be designed to exhibit a broader range of (e.g., mechanical) properties rather than a particular targeted value. Therefore, the range of achievable properties can be expanded using random networks and may realise smooth variations in properties. An example is the rational design of microstructures to regulate elastic mechanical properties separately (i.e., the duo of elastic stiffness and Poisson’s ratio) [31,45]. The theoretical upper limits for the mechanical properties of lattices in 2D or 3D have been defined by Hashin and Shtrikman [46]. It has been observed that the application of lattices with anisotropic microstructures can enhance these theoretical upper bounds [47]. The second advantage is that random networks are less susceptible to local defects created during the AM process due to their stochastic nature. Third, their design process is much more straightforward than that for uniform and ordered networks. In ordered networks, the structural integrity and assembly of unit cells are fairly challenging tasks. In contrast, it is easier to combine several types of unit cells in random network lattices, such as combining stretch-dominated unit cells with bending-dominated unit cells.

### 2.2. Topology Optimisation Designs

Topology optimisation (TO) can be defined as the application of mathematical models to design optimised arrangements of microstructures of porous structures to obtain desired and optimum properties while satisfying certain conditions. TO algorithms combined with computational models help designers to determine topologically optimised constructs as well as local microstructural compatibility [32]. Several optimisation approaches have rapidly evolved and been applied for this purpose in AM [48], among which “inverse homogenisation” is an example [49,50]. TO using homogenisation methods provides tools to realise targeted effective and unusual properties through the disposition of unit cells and material distribution in 3D space. Examples of these atypical properties are the negative thermal expansion coefficient [51] and the negative refraction index [52].

Various objective functions can be considered for the design of AM lattices. An example of an objective function can be defined based on maximising the specific stiffness (i.e., stiffness-to-mass ratio), which can lead to lattices with similar anisotropic spongy-bone microarchitectures [53]. There are some optimisation models that have been developed by considering bone tissue adaptation processes [11,54,55] in order to create the optimal designs of microstructures of lattice parts that are often used for the creation of bone scaffolds and orthopaedic implants in biomedical engineering (Figure 1d) [56,57,58,59]. Strain energy can also be defined as another objective function for the TO of load-bearing lattice structures.

For multi-physics optimisation problems, the TO of lattice structures can be defined such that multiple objective functions can be optimised [52]. This allows for the production of materials with multi-functional properties. Examples include the design of lattice geometries with two combined mutually exclusive properties, such as a maximised bulk modulus or elastic stiffness and permeability [60,61]. This can also be performed using the TO of functionally graded porous biomaterials [62].

Several optimisation techniques have already been developed and applied in the design of optimised topologies for lattice structures with multi-functional properties. These include evolutionary structural optimisation [63,64], solid isotropic materials with the penalisation method [65,66,67], the bi-directional evolutionary structural optimisation method [68,69], and level-set algorithms [70]. There are various commercial optimisation tools (e.g., TOSCA, Pareto works, and PLATO [71]) and free codes [71] available for the TO of AM lattices.

Current research integrates the design aspects of TO with AM fabrication features [72,73], such as the procedure that deals with optimising the disposition of support materials during the AM process. This integration helps alleviate stress concentrations at struts and their junctions in lattice structures during or after 3D printing, when the support materials are being removed, thus saving material and shortening the lead time [16,74].

### 2.3. Bio-Inspired Design

Another approach in the design of lattice structures is bio-inspired design. Natural cellular materials, such as bone, cork, and wood, can enrich scaffold design libraries [75,76,77]. Various key design elements present in the structures of natural materials (e.g., functional gradient and hierarchy) can be translated into bio-inspired porous materials, primarily for biomaterials employed in tissue engineering. An evident instance of natural cellular material is cancellous or trabecular bone—a porous biological material mainly composed of hydroxyapatite minerals and collagens shaped at several hierarchical levels. A connected network of trabeculae in the form of rods and plates forms the cellular structure of cancellous bone [78]. The distribution of trabecular microstructures is a functionally graded placement where the porosity close to the outer shell is lower than that of the inner shell of the bone. The design of bio-inspired lattice structures can benefit from mimicking these features (Figure 1e). Co-continuous multi-material cellular constructs with inter-penetrated boundary phases exhibit multi-functionality and remarkable mechanical properties, such as gradient stiffness in one layout (Figure 1f) [79]. In this respect, AM technologies can create such components with smooth transitions of target parameters in three dimensions and minimise stress concentrations at interfaces [33,80,81,82].

The importance of this aspect becomes more visible for orthopaedic implants used to treat large bone defects when the bone cannot go through the natural self-healing process. In such cases, external intervention is necessary to facilitate the healing process [9,83], but the repair can be challenging. The optimal biological choice is the use of either autograft (tissue taken from the patient) or allograft (tissue taken from another donor or person) [84]. However, these methods can lead to several secondary issues, such as problems with harvesting tissue from the patient or the risk of transmitting diseases between patients in the case of allograft tissue. The alternative solution is to design and implant biomimetic materials and constructs to repair skeletal defects.

One method of establishing the geometry of biomimetic lattice constructs is to derive the original configuration by using non-destructive imaging methods, such as computed tomography (CT) or magnetic resonance imaging (MRI). Image-based design methods have been extensively used to design implants and bio-prostheses in tissue reconstruction applications [85]. These non-destructive imaging modalities have also been used to determine the shape variations of long bones at different anatomical locations [86]. Another significant advantage of using the imaging method is the possibility of developing patient-specific implants, where the geometry of the implant is based on the configuration of the target bone of the individual [87,88,89].

### 2.4. Meta-Biomaterials

“Batch-size-indifference” and “complexity-for-free” are two additional characteristics of design for AM [11,90]. These features have flourished in the creation of patient-specific meta-biomaterial implants with tailored properties using “designer material”. Designer materials, also known as mechanical metamaterials, are defined as advanced engineering materials that exhibit remarkable properties based on their microarchitectural designs rather than their chemical compositions [91,92]. One of these atypical characteristics is the negative Poisson’s ratio or auxetic property [93], which is defined as a lateral expansion upon longitudinal extension. Penta-mode metamaterials [94], shape matching [95,96,97], rate dependency [98,99], crumpling [100], and action-at-a-distance [101] are other examples of these unusual properties that can be achieved by the rational design of engineered mechanical metamaterials. Three major types of unit cells with auxetic properties can be identified, namely, re-entrant, chiral, and rotating (semi-)rigid (Figure 1g) [34]. These designs have been implemented and additively manufactured in 2D or 3D. Among the abovementioned designs, the re-entrant unit cell is one of the most straightforward designs that enables the control of the values of Poisson’s ratio by merely changing the angle of struts. It is also the more researched type of unit cells with auxetic properties as compared to the other designs.

There are reports on auxetic behaviour in skeletal tissues, such as tendons [102] and trabecular bone. It has been observed that scaffolds with auxetic properties promote neural differentiation. This can be attributed to them providing mechanical cues to pluripotent stem cells [103]. There is not much evidence on the advantages of auxetic behaviour in improving bone tissue regeneration thus far. Nevertheless, it has been reported that the hybrid design of meta-biomaterials (i.e., the rational combination of unit cells with positive and negative values of Poisson’s ratio) enhances the longevity of orthopaedic implants [104]. As evidence, it has been observed that the hybrid design of meta-biomaterials for the hip stem prevents the development of a weak interface between the implant and bone and, consequently, prevents the loosening of the implant. This is particularly important because wear particles released by implant loosening can cause inflammatory responses in the body [105,106,107]. Additionally, auxetic meta-biomaterials exhibit superior quasi-static [108] and fatigue performance [35], enabling them to be good candidates for load-bearing (e.g., hip stems) applications. The surface and under-structure of meta-biomaterials can also be engineered using post-processing techniques, such as abrasive polishing, electropolishing [109], and hot isostatic pressing [110], which can improve their surface finish and mechanical properties.

Other geometrical designs with non-auxetic properties (cube, diamond, rhombic dodecahedron, etc. [111]) have also been explored for use in biomedical devices, such as space-filling scaffolds (Figure 1h) [36].

Owing to the unique features of TPMS-based porous structures, these geometries are immensely popular as designs for meta-biomaterials [30,112,113,114,115]. First, their mean surface curvature is fairly similar to that of trabecular bone [116,117,118]. Second, the importance of the surface curvature as a mechanical cue in tissue regeneration has been reported [8,119,120,121] and extensively discussed in several studies [29]. Therefore, it can be assumed that TPMS-based porous meta-biomaterials may enhance tissue regeneration performance. It has also been reported that TPMS-based geometries can provide a perfect balance between mechanical properties (i.e., elastic modulus and yield stress) and mass transport characteristics (i.e., permeability) [30,122] and achieve a balance similar to that of bone. The multi-physics properties of TMPS-based geometries can also be decoupled by combining multi-material 3D printing and parametric designs using mathematical approaches (e.g., hyperbolic tiling) [123].

Different forms of 2D and 3D shape-shifting mechanism-based designs (e.g., multi-stability [124] or self-folding techniques using the origami or kirigami approach [125,126]) have also been employed to create advanced meta-bioimplants with enhanced properties and functionalities (Figure 1i). Examples are deployable meta-bioimplants [127,128] and 3D foldable curved-sheet (i.e., TPMS) lattices made with origami-folding techniques [129]. One of the benefits of the transition between (2D) flat constructs to 3D meta-biomaterials is that, in such cases, the surfaces can be decorated with additional functionalities. Examples of such induced features are nano-patterns (Figure 1j) [37].

Kinematic or compliant mechanisms can also be employed in the design of meta-biomaterials. This allows for fabricating non-assembly mechanisms with compliant or rigid joints [130]. Non-assembly designs have shown great potential in the fabrication of orthopaedic implants using shape-morphing metallic clays [131].

## 3. AM of Biomedical Metals and Alloys

There are many areas in which metals and their alloys can be used in biomedical applications. Upon their contact with the biological environment, most metals undergo corrosion and ion release, which may be harmful to the body. Therefore, they must show an excellent biocompatibility response in vivo [132]. Titanium (Ti) and most of its alloys, stainless steel, cobalt (Co)-based alloys (such as CoCrMo), zirconium (Zr), niobium (Nb), and tantalum (Ta) are some examples of biocompatible metals and alloys. They exhibit magnificent corrosion resistance and good mechanical properties and are excellent biocompatible materials [133].

Among various biocompatible metals and alloys, Ti and its alloys (e.g., Ti6Al4V) are probably the most extensively studied materials [25]. Ti6Al4V is relatively inexpensive and has lower ductility than pure Ti. However, pure Ti with lower mechanical strength but higher ductility is considered a highly biocompatible metal. Stainless steel, while being cheaper than others, is relatively biocompatible. Laser powder bed fusion (L-PBF) processes can easily manufacture stainless steel, and its elastic modulus is higher than that of Ti6Al4V [25]. Ti6Al4V exhibits appropriate fatigue behaviour in terms of fatigue strength, but its fatigue strength is lower in comparison to some other metallic materials, such as CoCr [134].

Biomedical metals and alloys are good candidates for use as porous implants in orthopaedic applications (Figure 2a,b). However, their elastic moduli are significantly larger than those of the replacing bones. To prevent stress shielding from occurring at the bone–implant interface, the elastic modulus and yield strength of metallic implants must be tuned accordingly. Several methods can enhance the mechanical properties of bone and metal interfaces, such as creating graded metallic porous implants. Another feasible approach is to introduce certain elements to the structure of the alloys, which reduces the elastic moduli of porous structures; for example, adding ß-phase-stabilising elements (e.g., Ta, Nb, Zr, and Mo) to Ti can create ß-type Ti alloys with lower elastic moduli as compared to Ti6Al4V. Examples of such ß-type Ti alloys that improve the mechanical compatibility of implants are Ti13Nb13Zr (with an elastic modulus of 79 GPa) [135] and Ti29Nb13Ta4.6Zr (with an elastic modulus of 55–65 GPa) [136].

Surface treatments and coatings can improve the performance of metallic implants in regenerating bone tissue (Figure 2c–e) [137,138,139,140,141,142,143,144]. It has been reported that surface modification processes, such as introducing bioactive glass and mesoporous bioactive glass to the surfaces of Ti6Al4V scaffolds [140], can enhance the bone tissue regeneration performance. Furthermore, surface biofunctionalisation processes using plasma electrolytic oxidation (PEO) [141] with or without silver, zinc, or copper nanoparticles can have potential immunomodulatory effects and can minimise implant-associated infections (Figure 2d) [138,142,143]. Furthermore, the bactericidal and osteogenic performance of metallic implants can be controlled by decorating their surfaces with nanostructures. An example is using inductively coupled plasma reactive ion etching to fabricate Ti nanostructures [144]. Layer-by-layer coating biofunctionalisation is another approach to impart multiple functionalities simultaneously (e.g., improved tissue growth factors as well as antibacterial behaviour) to metallic (e.g., pure titanium [139]) implants (Figure 2e).

### 3.1. Biodegradable Metals

Biodegradable materials used for biomedical implants are defined as materials that can gradually degrade in the human body over time. They can be either polymer-based or metal-based [146] biomaterials. The primary function of biodegradable metals is to be temporarily present in the body to assist in the healing process and vanish following its completion. The parts and products of biodegradable metals may be fabricated utilising AM techniques. Some examples include pure iron [147] and magnesium alloy (WE43) [148] porous structures. Many medical devices and implants may benefit from biodegradable metals, such as Mg alloys that have already been used as biodegradable materials for cardiovascular stents [149] and bone screws [150]. Fe-Mn-Si alloys, such as the alloy with about 30% mass Mn and 6% mass Si [151], were found to exhibit the shape-memory effect, which looks quite promising for medical and other industrial applications. Martensitic transformation also enhances the mechanical properties of alloys, such as hardness, strength, and fatigue resistance [152].

The rate of biodegradation or bio-absorbability of biodegradable metallic implants in the body is a crucial parameter. For example, the degradation rate of Zn-based alloys, which are known as one of the most suitable biodegradable metals, is around 20–300 μm/y in vitro [153,154], while for Fe- and Mg-based alloys, this rate is lower than 50 μm/y and higher than 300 μm/y, respectively, in in vitro conditions [155,156]. The degradation rate of pure Mg is the highest when it comes in contact with the chloride-containing physiological environment. Hydrogen gas is produced at a high rate by the corrosion of Mg, which cannot be managed inside the host body. However, the degradation rate of Fe-based biodegradable metals is much slower. Alloying has been recognised as an effective way to tune the biodegradation rate. Mg-based alloys with elements such as Y, Sr, Zn, Zr, and Ca have exhibited significantly lower biodegradation rates as compared to pure Mg. These alloys also exhibit good strength properties, making them suitable for manufacturing load-bearing implants or implant components [157].

Apart from alloying, the biodegradation rate can also be regulated by increasing the surface area. Therefore, two practical tools that can be used to manipulate the degradation rates of such materials are the geometry and level of porosity. In addition to the effect of environmental conditions, other physical conditions, such as cyclic mechanical loading, can increase the biodegradation rate of Mg alloy (WE43) [158], porous iron [159], or zinc [160] scaffolds.

The AM fabrication process for porous biomaterials using biodegradable metals is considerably challenging, particularly in the case of Mg and its alloys, which have high flammability, strong chemical activity, low melting points, and low evaporation temperatures. For some Mg alloys, there is the potential of developing crystallisation cracks because of fusible eutectics, great deformation, and stresses due to a high linear thermal expansion coefficient and a broad range of crystallisation temperatures [161]. Their fabrication thus requires special safety precautions and process modifications.

Another approach to creating biodegradable porous metals is to use extrusion-based AM techniques (Figure 2f). For such techniques, it is necessary to create an ink formulation that matches the 3D printing process as well as the debinding and sintering steps [145,162]. In vitro corrosion results showed an improvement in 3D-printed iron scaffolds compared to bulk materials [145]. This can even be more controlled by creating functionally graded biodegradable porous metals (e.g., iron [163] and zinc [164]).

It is also notable that employing non-biodegradable materials for implants may terminate natural bone ingrowth, which may require subsequent surgery to facilitate further bone growth. Therefore, biodegradable materials are a better option for many implant applications. However, a critical concern regarding biodegradable materials is the cytotoxicity phenomenon that arises from the biodegradation process within the body of the patient [25].

### 3.2. Shape-Memory Alloys

Shape-memory materials have the ability to return from a deformed state (temporary shape) to their original (permanent) shape when provoked by external stimuli [165]. This effect arises from the temperature-driven phase transformation of shape-memory alloys (SMAs). SMAs have recently gained increasing popularity for their use in orthopaedic implants as well as cardiovascular devices. A typical SMA is Nitinol (NiTi), which comprises equal atomic percentages of Ni and Ti. The shape-memory effect of NiTi emerges from the change from austenite to martensite at high and low temperatures, respectively [166,167].

Bulk NiTi with an elastic modulus of approximately 48 GPa, which is significantly lower than those of Ti alloys, can recover relatively large strains of up to 8%. It is pseudoelastic, which implies that NiTi is capable of recovering large strains upon unloading at a constant temperature [168,169]. These properties make NiTi a suitable candidate for the manufacturing of many medical devices, including surgical guides, stents, orthodontic wires, plates, and staples for bone fracture healing purposes.

The lattice structures of an approximately equiatomic Ni-Ti alloy are recognised as favourable bioimplants and biological micro-electro-mechanical systems (bio-MEMS). This can be attributed to their unique combination of thermal and mechanical shape memories, which is based on the reversible martensitic phase as well as high corrosion resistance, superelasticity, and biocompatibility properties [167,170]. Because Ni is highly allergenic, its presence in NiTi may raise concerns regarding the biomedical applications of Nitinol [171,172]. Some surface modification techniques or element replacement may be required to alleviate this effect while maintaining biocompatibility (Figure 2c) [137,173]. For example, TiNb and other developed alloys (i.e., TiNbX, where X = Zr, Ta, or Hf) exhibit elastic strains of up to 4.2% [174].

The potential applications of SMAs include deployable orthopaedic implants [127,128] and 4D-printed implants (i.e., implants with 3D-printed structures whose properties change over time) (Figure 2g) [96,175,176,177].

According to the findings of Tsaturyants et al. [178], a combination of thermal cycling and heat treatment can decrease the temperature range of martensitic transformation and also greatly enhance the mechanical properties of the Nitinol alloy processed by L-PBF. They concluded that a combination of heating–cooling cycles of 350 and 400 °C over the temperature range of martensitic transformation can result in a 10 to 15 °C decrease in the martensitic transformation temperature and can also add another step to the transformation sequence of the structure. They also observed a ~7% increase in the maximum stress and dislocation yield stress, as well as a ~10% increase in the difference between the dislocation and transformation yield stresses of the developed structure, by applying 10-cycle heating–cooling.

### 3.3. In Situ Alloying and Composites

The capability of dispensing materials within 3D lattice structures and placing several materials in desired positions within the entire structure is granted with AM technologies. Such a capability increases the design complexity, particularly when there is already a need for the intricate geometry design of lattice structures.

In situ alloying is defined as the process of combining several feedstock materials with different compositions and simultaneously feeding them into the melt pool. Such a compositional mixture can attain customised properties and functionalities [179]. SLM-processed in situ Ti-26Nb alloy for biomedical applications is an example of a compositional mixture [180].

Generally, adding reinforcing particles (mostly ceramics) and in situ alloying to metal matrices can greatly enhance the mechanical characteristics, such as hardness, stiffness, and strength. Combining them with metal also affects the intrinsic properties, including toughness and/or electrical/thermal conductivity. Reinforcing particles can be introduced through ex situ mixing methods, such as ball milling, or formed in situ during AM processes by combining metal matrix and alloying elements or ceramic reinforcing particles. Laser power and other process parameters of the L-PBF process may be tuned to ensure complete melting of the metal matrix and alloying elements for full interaction with the surrounding ex situ particles as well as a maximum response between the matrix and alloying elements [181]. Under dedicated L-PBF process conditions, a Ti-TiB porous composite was created through an in situ reaction between the Ti matrix and TiB_2_ reinforcing particles [182]. However, owing to several factors, such as weak interfacial bonding, incomplete reactions, interfacial cracks, and inhomogeneous dispersion of the added particles, it is difficult to create a perfect composite lattice structure. Metallic porous composites are not limited to ex situ or in situ composites. It is also possible to fabricate porous metallic glass composites by using L-PBF methods, in which the reinforcing agents are generally crystalline phases distributed in the porous amorphous matrix [183].

The capability of L-PBF processes to produce various metal matrix composites has already been demonstrated. Distinct advantages can provide benefits for the production of desired parts [162,184]. These include the ability to build cellular structures that are reinforced by composite components at desired locations. Such advantages have provided researchers with opportunities to introduce lattice or non-lattice structures built on functionally graded materials (FGM) for biomedical applications, which can be considered for further investigation and exploration [162,184].

## 4. AM of Biomedical Polymers

Polymers were the first materials used in AM. Their lower melting points, compared to ceramics and metals, as well as their modifiable chemical structures, make them suitable for manufacture using AM technologies, such as material extrusion, powder bed fusion (PBF), and vat photopolymerisation [185,186].

In addition to possessing properties suited to manufacturing, polymers for biomedical applications should be compatible with the host tissue and degrade after tissue regeneration. As a result, polymers require other properties, such as biocompatibility and biodegradability, to be suitable for implants and other biomedical applications related to natural tissue regeneration, where the polymer is intended to be replaced by the tissue [187]. Polymers with such characteristics can be broadly classified as natural and synthetic polymers.

Synthetic polymers are more hydrophobic and mechanically more stable than natural polymers due to their slower degradation rates. On the contrary, faster degradation, which may result in lower mechanical strength over time, is ideal for tissue regeneration, as the persistence of biomaterials implanted in the host tissue may trigger physical impairment [187]. Furthermore, the fatigue behaviour of 3D-printed polymeric materials is also of great importance for medical devices [188,189]. Therefore, the choice of materials and their combinations to obtain properties suitable for targeted medical devices is challenging.

### 4.1. Hydrogels

When considering the biomedical applications of polymers, it is necessary to include hydrogel, a new and promising polymeric material with a substantial role in various aspects of healthcare and biomedical engineering. Hydrogels are described as three-dimensional crosslinked polymer networks that are able to absorb and retain a large quantity of water [190,191,192,193]. Owing to several important properties, such as hydrophilicity, biocompatibility, and nontoxicity [194,195,196], hydrogels have been instrumental in tissue engineering and pharmaceutical applications, including drug delivery, wound healing dressings, and in vitro cell culturing [191,197]. Hydrogels are suitable for extrusion-based bioprinting because of their non-Newtonian shear thinning behaviour, but there are some limitations in terms of the printing characteristics [198]. Following extrusion-based 3D printing and before crosslink formation, hydrogels have poor shape fidelity, limiting their capacity to form larger structures [198,199,200]. In recent years, new techniques for cell-seeded biofabrication and novel bio-inks have been developed to overcome this shortcoming [201].

Hydrogels show viscoelastic behaviour, and therefore, their rheological properties are of great importance, as they can determine the success of the 3D printability of these materials [202,203,204]. The rheological properties of hydrogels originate from their microstructures and can provide information on the rate and nature of deformation under imposed strain or stress. There are several procedures to control molecular structures and, consequently, the rheological properties of hydrogels. Examples of these procedures are chemical (e.g., water and/or other solvents) and physical (e.g., UV irradiation) crosslinking, which can be used to tune the elastic properties of hydrogels [205,206].

The main advantages of hydrogels include their biocompatibility, better encapsulation, growth, and protection of cells and fragile drugs due to their high water content, modifiable mechanical characteristics as a result of crosslinking, better transfer of nutrients to cells and waste products from cells, controllable drug release, and the simplicity of patterning using 3D printing [190,197]. Their limitations include difficulties in physically manipulating structures, restricted use in load-bearing applications due to their poor mechanical properties, time-consuming printing optimisation, and difficult sterilisation [190,197].

Three types of hydrogels can be realised and classified based on the origin of their polymers, namely, natural, synthetic, and synthetic–natural or hybrid hydrogels [207].

Anionic polymers, such as hyaluronic acid (HA), alginic acid, carrageenan, pectin, chondroitin sulphate, dextron sulphate [190], cationic polymers (such as chitosan and polylysine [191]), natural polymers (such as agarose), and amphipathic polymers (such as collagen, fibrin and carboxymethyl chitin [197,208]), are just a few examples of a wide range of natural biodegradable polymers and their derivatives that produce hydrogels. Moreover, synthetic polymers can also be used to create hydrogels. Examples are polyacrylamide (PAAM), polyethylene glycol (PEG), and polyvinyl alcohol (PVA). Recently, synthetic polymers have gained popularity over natural polymers owing to their higher water absorption capacity, better mechanical strength, slower degradation, and durability [209,210].

Hydrogels can also be classified according to their polymeric composition and preparation method. The first examples are homopolymeric hydrogels, which are composed of a single structural unit derived from a sole type or monomer [211]. Second, copolymeric hydrogels are formed from two or more species and at least one hydrophilic constituent ordered in an irregular or interchanging configuration within the chain of the polymer network [212]. Third, multipolymer interpenetrating polymeric network (IPN) hydrogels are composed of two independent crosslinked natural and/or synthetic components, forming a network [213,214].

### 4.2. Natural Polymers (Hydrogel)

Many natural biopolymer hydrogels, such as alginate, cellulose, agarose, fibrin, chitosan, gelatine, hyaluronic acid, and gellan gum, have already been employed in bio-printing applications [215]. Natural polymer hydrogels, such as gelatine, chitosan, alginate, and collagen, have been used to repair biological tissues, including bone, nerve, cartilage, and skin [216]. They usually have good biocompatibility and cause minimal inflammatory and immunological responses in the host tissue. Furthermore, they have been evaluated for use as scaffolds in tissue engineering because they are naturally biodegradable, in addition to being biocompatible and possessing vital biological functions; however, most natural polymers do not meet clinical requirements owing to concerns about potential immunogenic reactions as well as relatively low strength and toughness [217,218]. Chemical and physical modifications or other processes, such as compositing and introducing micro- or nano-structures, can be utilised to impart specific functionalities and improve these deficiencies [207].

Natural polymers are classified into four categories: proteins, polysaccharides, protein–polysaccharide hybrid polymers, and polynucleotides [219]. The first category (i.e., proteins) includes collagen, fibrin, gelatine, silk, lysozyme, and genetically engineered proteins (such as calmodulin, elastin-like polypeptides, and leucine zipper) [216,217,218]. HA, chitosan, dextran, and agarose belong to the second category (i.e., polysaccharides) [220,221]. Collagen–HA, gelatine–chitosan, laminin–cellulose, and fibrin–alginate are examples of the third category, which is a hybrid of proteins and polysaccharides [222]. Finally, polynucleotides include DNA and RNA [223]. In this review, we focus on the three most commonly used classes of hydrogels, namely, collagen, gelatine, and alginate. More information on different classes of natural hydrogels can be found extensively in previous studies [207,224,225,226].

#### 4.2.1. Collagen

Collagen, a vital component of the extracellular matrix (ECM) that regulates cell functions and mimics tissue characteristics [227], is a popular biopolymer in AM [186,228]. Material extrusion is the more common method of manufacturing 3D collagen structures in comparison to powder bed techniques because collagen denatures at high temperatures, and good flowability of the powder bed cannot be achieved [229,230].

Owing to its outstanding biological features, such as good biodegradability, cell adaptability, and antigenicity, many applications can be identified for collagen in tissue engineering and drug delivery systems [231]. However, the degradation rate and mechanical properties of natural collagen are not adequate for tissue engineering purposes. For instance, the elastic modulus of atelopeptide collagen hydrogel (Type I Collagen) is approximately 65.5 KPa [232], which is much lower than that of the actual articular cartilage. Some modifications, such as crosslinking or mixing with other materials, can be applied to improve these properties and make them suitable for specific applications in tissue regeneration [226,233].

Several factors, such as the collagen concentration and the variety of crosslink, can define the resulting microstructure and mechanical properties of collagen-based scaffolds [234]. For example, if the genipin percentage is approximately 0.1%, there is no significant change in porosity. A higher concentration, however, causes a decrease in the porosity of the scaffold [234].

In many tissue engineering applications and wound healing, the combination of collagen with other materials has led to the enhancement of its properties [235]. For example, the combination of synthetic polymers with collagen improves its mechanical strength, and its combination with growth factors modifies its regeneration behaviour in tissue engineering applications [236].

#### 4.2.2. Gelatine

Gelatine is another natural biopolymer derived from animal by-products, such as bones, connective tissues, and skin. Gelatine is popular because it is inexpensive and has desirable biological properties (e.g., biocompatibility, biodegradability, and non-immunogenicity [237,238]). Gelatine can provide an appropriate structure and necessary nutrients for the growth and distribution of cells. The most common gelatine application is the formation of hydrogels and vessels for controlled drug release [237,239].

A novel method for organ and tissue printing on a gelatine matrix is based on the presence of hepatocytes [240]. Hepatocytes are the primary epithelial cells of the liver, which maintain their morphology in culture dishes coated with ECM components [240]. The printing of gelatine constructs can be performed using the extrusion method with hepatocytes at a lateral resolution of 10 µm, allowing hepatocytes to remain viable for approximately two months [241].

Gelatine in its unmodified form experiences sol–gel transition, but the gelation speed is slow, which cannot ensure the exactness of the construct formation. Gelatine methacrylate (GelMA) may be used to speed up the process and overcome this shortcoming [242]. This relatively inexpensive solution results in good biocompatibility and biodegradability [243,244]. The combination of GelMA and methacrylate polyvinyl alcohol can be used in the presence of a visible light photo-initiator to generate a bio-resin for digital light processing lithography [215]. Freeform fabrication without the generation of lattices is possible with a small resolution of 20–50 µm by applying this method [215].

#### 4.2.3. Alginate

Alginate is an important hydrogel that can be obtained from brown algae and has wide applications in tissue engineering, drug delivery, wound healing, and bioprinting [191,208,245,246]. It is formed from blocks of (1, 4)-linked β-d-mannuronate (M) and α-l-guluronate (G) residues [191]. The three factors that affect the physical properties of alginate are its composition (i.e., M/G ratio), G-block length, and molecular weight [247]. Increases in the length of the G-block and molecular weight enhance the mechanical properties of alginate [191]. The gelation temperature, which influences the gelation rate, is essential for these properties. A gel can be formed as a result of the interaction between the carboxylic acid of alginate and bivalent counter ions, such as calcium ions (Ca^2+^) [248]. At lower temperatures, the reactivity of ionic crosslinkers (i.e., calcium ions) decreases, resulting in slower crosslinking and a more ordered network structure [249].

The popularity of alginate as a biomaterial for biomedical applications stems from its easy and fast gelation, low cost, and lack of immunogenicity [250]. Printability is another advantage of alginate-based hydrogels. This indicates that printing capabilities can be easily modified by providing different polymer densities or adding calcium chloride to change the crosslink [251,252,253]. Their mechanical properties are also tuneable, implying that they can be adjusted to improve printability and accuracy [192].

Utilising various crosslinkers and combining other polymers, such as gelatine, can rectify the mechanical properties as well as the cell affinity of alginate because alginate does not provide sufficient cell attachment and proliferation [254]. Thus, the biocompatibility and support of cellular function and differentiation in alginate and the good cell attachment characteristics of gelatine can be achieved [216]. Therefore, alginate–gelatine with excellent rheological properties has been introduced in various biomedical applications [216]. This alteration in rheological properties also changes the viscosity [251] of the hydrogel and makes it suitable for extrusion-based 3D printing.

### 4.3. Synthetic Polymers

Synthetic polymers, such as synthetic hydrogels and thermoplastics, have been used in 3D printing processes for considerable time. Synthetic polymers have higher mechanical strength, a better controlled degradation rate, and improved processability compared to natural polymers. Their low thermal expansion coefficient, glass transition temperature, and melting point compared to natural polymers make them suitable for desired applications.

However, robust secondary bonding is still required for the best resulting strength, as a 3D printing procedure involves the layer-by-layer addition of materials. Although PMMA (i.e., polymethyl methacrylate) has many favourable characteristics for use in medical applications, such as medicine, denture bases, filling of bone and skull defects, bone implant fixation screws, and vertebrae stabilisation, it is not widely employed in 3D printing due to the poor bonding between 3D-printed PMMA and the build plate as well as metals [255]. It requires a higher temperature, is susceptible to warp and distortion, needs glue to adhere to the bed, and requires a bed temperature of at least 60 °C [256]. In a study on 3D-printed PMMA, infiltration with epoxy was applied to increase the tensile strength and elastic modulus of the printed part from 2.91 MPa and 223 MPa to 26.6 MPa and 1190 MPa, respectively [257]. In addition, infiltration with wax was shown to improve the surface quality of the part [257].

In AM for biofabrication, direct printing of a cell-seeded material or “bio-ink” can be clearly distinguished from the printing of a cell-free scaffold with a “biomaterial ink” that can be directly implanted or seeded with the cells afterwards [217]. Bio-inks are generally produced from hydrogels, which are very well established as suitable materials for 3D cell cultures. They also have excellent biocompatibility and highly adaptive physical, mechanical, and biological properties [194,258,259].

Biomaterial inks composed of thermoplastics, ceramics, composites, and metals are often used to provide a rigid scaffold for the permanent or slow-degrading stabilisation of a construct, while bio-inks can provide a much softer scaffold, and the deposition of a new ECM can be replaced more quickly by the embedded cell population [198,217,260].

#### 4.3.1. Synthetic Hydrogels

Synthetic hydrogels can be easily synthesised and manipulated together on a large scale at a molecular level by polymerisation, crosslinking, and functionalisation [261]. However, the majority of them only function as passive scaffolds for cells. They do not promote any active cellular interactions by themselves. Natural polymers, including proteins, have different structures and are involved in the regulation of active cellular responses, biological recognition, and cell-triggered remodelling. Consequently, combining the properties of synthetic and natural polymers to create hybrid hydrogels has developed into a direct method of developing bioactive hydrogel scaffolds for tissue engineering [219].

Three primary classes of synthetic polymers are recognised for creating synthetic hydrogels, namely, non-biodegradable, biodegradable, and bioactive polymers [219]. Tissue engineering applications of non-biodegradable hydrogels primarily involve bone and cartilage [262], with relatively limited applications in vascular constructs or other soft tissues. For these applications, maintaining physical and mechanical integrity is essential for the hydrogel. A vital consideration in the scaffold design is the mechanical stability of the gel, which can be enhanced by introducing crosslinking components and comonomers and by modifying the level of crosslinking [263,264,265]. A much higher degree of crosslinking can also result in brittleness and decreased elasticity, and therefore, the optimal degree of crosslinking must be identified.

In order to provide the desired flexibility of the crosslinked chains and facilitate the movement or diffusion of the incorporated bioactive agent, an adequate elasticity of the gel is required. Therefore, there is a need to compromise between mechanical strength and flexibility by selecting the best components and percentages in the construction of non-biodegradable hydrogels as tissue-engineering scaffolds [219].

Copolymerisation of different vinylated monomers or macromers, such as 2-hydroxyethyl methacrylate (HEMA) and 2-hydroxypropyl methacrylate (HPMA), can produce non-biodegradable synthetic hydrogels [262,266,267,268]. Another method to generate non-biodegradable hydrogels is to use non-biodegradable polymers, such as modified polyvinyl alcohol (PVA) and PEG [269,270,271].

PEG has several unique properties, such as solubility in water and organic solvents, nontoxicity, moderate protein adherence, and no immunogenicity. These properties make PEG the most widely investigated polymer for creating hydrogels [262,268]. Another synthetic hydrophilic polymer that can be mixed with other water-soluble polymers to create hydrogels in tissue-engineering applications is PVA [270,271].

An essential consideration in the construction of scaffolds for tissue engineering is biodegradability. The desirable rate ensures that biodegradation corresponds to new tissue regeneration at the corresponding site [272,273,274].

The most widely used biodegradable polymers for scaffold fabrication are polyesters, including polylactic acid (PLA), polyglycolic acid (PGA), polycaprolactone (PCL), and their copolymers [273,275]. They can be employed to improve hydrophilic polymers, such as PEG, to develop acrylate macromers or amphiphilic polymers, and to produce biodegradable hydrogels via chemical or physical crosslinking [276,277,278,279,280,281,282,283,284,285,286,287]. A lack of cell-specific bioactivities, such as cell adhesion, migration, and cell-mediated biodegradation, is the major limitation in their use as tissue-engineering scaffolds. These limitations can be alleviated by introducing bioactive molecules into the synthetic hydrogels [272,288,289,290]. Bioactive elements can be attached to the hydrogel network, such as peptides, during or after hydrogel formation [217,258]. Different ECM component-derived peptides or bioactive molecules, such as cell-adhesive [259,260] and enzyme-sensitive [291,292], have been used to modify synthetic polymers for fabricating bioactive hydrogels.

Physical properties (e.g., network parameters and diffusive profile), mechanical strength, and biological properties (e.g., cell adhesion, migration, and scaffold biodegradation) can be engineered using molecular design [219]. Unlike natural hydrogels, bioactive synthetic hydrogels offer much broader control to improve the matrix architecture and chemical composition and provide a biomimetic environment for tissue regeneration and cell growth.

#### 4.3.2. Polylactic Acid (PLA)

PLA can be produced from renewable resources [293]. Its biocompatibility, biodegradability, and bioresorbability have made it a suitable candidate for a broad range of biomedical applications, including neural and vascular regeneration [294], stents [295,296,297], surgical sutures [298], plates and screws for craniomaxillofacial bone fixation [299], interference screws in the ankle, knee, and hand, tacks and pins for ligament attachment, anchors [300], spinal cages [295,301], soft-tissue implants, tissue-engineering scaffolds, tissue cultures, drug delivery devices [302], and craniofacial augmentations in plastic surgery [303].

PLA can be produced using various polymerisation methods from lactic acid, including polycondensation, ring-opening polymerisation, and direct processes, such as azeotropic dehydration and enzymatic polymerisation [304]. Compared to other biopolymers, there are numerous advantages associated with the production of PLA: (*i*) Eco-friendly: it can be obtained from renewable resources in nature (e.g., corn, wheat, or rice). PLA is biodegradable, recyclable, and compostable [305,306], and its production consumes carbon dioxide [307]. (*ii*) Biocompatibility: biocompatibility is undoubtedly the most important aspect of PLA, particularly with respect to biomedical applications. (*iii*) Processability: thermal processing of PLA is easier compared to that of other biopolymers, such as polyhydroxy alkanoate (PHA), PEG, and PCL.

Some of the shortcomings of PLA can be listed as: (*i*) Insufficient toughness: PLA is a brittle material with less than 10% elongation at the breaking point. (*ii*) Low degradation rate: the degradation rate of PLA depends on many factors, such as its crystallinity, molecular weight, distribution, morphology, water diffusion rate into the polymer, and stereoisomeric content. This feature leads to a prolonged in vivo lifetime, which in some cases can be up to 3 to 5 years [308]. (*iii*) Hydrophobicity: PLA is considered to be relatively hydrophobic. Its static water contact angle is assumed to be approximately 80 °C. This feature causes low cell affinity, and in some cases, inflammatory responses from the living host in direct contact with biological fluids have been observed [309]. (*iv*) Lack of reactive side-chain groups: PLA is chemically inert with no reactive side-chain groups, which results in an exciting approach towards surface and bulk improvements [310]. Considering the facts mentioned above, PLA bioactivity must be modified for its application in tissue engineering.

Several fabrication methods, such as particle/salt leaching [311,312,313], solvent casting [314,315], phase separation [316], gas foaming [317], freeze-drying [318], and electrospinning [319], have been employed to fabricate 3D scaffolds using PLA as the base material. Despite the successful manufacturing of scaffolds using these processes, these conventional methods have some drawbacks, namely, poor reproducibility, the use of toxic solvents, and limited control over the geometry of the scaffold and pores [320].

PLA-based scaffolds can also be formed through the SLA technique by copolymerisation of other materials, such as poly (d, l-lactide) and PEG, to achieve relatively good structures [321]. However, some limitations hinder the application of this technique for manufacturing PLA-based scaffolds, such as restrictions on the layer thickness and laser radiation to avoid over-curing or cytotoxic effects while using encapsulated cells [322]. Another disadvantage of this method is its high cost; furthermore, it is a more time-consuming process than other AM techniques.

PLA, unlike other biodegradable polymers (e.g., PCL), has limited use in SLS 3D scaffold manufacturing [323]. Because commercial PLA is typically available as millimetre-sized pellets, a process for developing particles with a smaller size prior to the SLS process is required to ensure the high resolution of 3D objects. Aside from the limitations on particle dimensions, the poor mechanical properties of sintered PLA scaffolds have also been reported [324,325].

The most common and cost-efficient technology for the 3D printing of PLA is Fused Deposition Modelling (FDM). PLA has appropriate thermal characteristics for FDM processing, which requires extrusion at temperatures ranging from 200 °C to 230 °C [326].

Process conditions and technical variables affect biocompatibility or accelerate polymer degradation, and they should be optimised such that the material does not experience excessively high shear stress during extrusion [322]. In addition, they affect the mechanical properties of additively manufactured PLA under static [327,328,329,330,331,332,333] and cyclic [334,335,336] loading. These process conditions can be identified as the thickness of layers (layer height), infill density, filling pattern, diameter and temperature of the nozzle, feed rate, printing speed, and build plate temperature; additionally, they exert a significant influence on the mechanical properties [337,338]. Increasing the layer height, for example, generates many voids in the microstructure of the printed part and reduces its tensile strength [339,340,341]. The tensile strengths and elastic moduli of 3D-printed parts are also affected by the extrusion temperature [342]. If the processing temperatures are too high, it can reduce the molecular weight of the polymer [343].

In addition to suitable mechanical properties, PLA-based scaffolds manufactured by the FDM method should also possess the desired biological properties to promote cell ingrowth. In this regard, the biocompatibility of PLA can be realised following the FDM process, ensuring that there is no cytotoxicity toward osteoblast-like cells [343]. It must also be determined whether the macro-patterns generated by the FDM equipment can induce cell differentiation and osteogenic processes [344].

Another consideration is that the hydrophobicity of PLA may limit its application in regenerative medicine, as it hinders cell adhesion and proliferation and the release of acidic by-products during the degradation process [345]. A promising approach is applying a bioactive coating on the surface of a 3D-printed PLA-based scaffold to improve its biofunctionality [346]. An alternative modification to enhance the biological properties of PLA is combining the base material with natural or ceramic additives [347]. A number of bioactive compounds for this purpose have been identified, such as chitosan [348], alginate [349], collagen [350], and calcium phosphates [351].

As an example, enhancement of stem cell adhesion, proliferation, and differentiation can result from coating the 3D-printed scaffold surface with polydopamine (PDA) [346,352,353] and acetylated collagen [354]. Ceramic additives can also be incorporated into the PLA matrix. This addition improves the hydrophilicity, osteoconductivity, mineralisation upon implantation, and mechanical properties of 3D structures [355,356]. An alternative approach is to mix the PLA matrix with natural polymers or their derivatives [357], such as o-carboxymethyl chitosan (CMC), which can substantially improve the hydrophilicity of the surface of the scaffold. The tensile modulus can also be increased with the controlled portion of CMC [357].

Applying a surface treatment to 3D objects to modify their topography or surface chemistry is another approach. The surface treatment can positively affect the attachment of cells and biological compounds to the structure [358,359]. Plasma treatment is one of the most investigated methods developed to enhance the surface chemistry of PLA-based parts without affecting their overall properties. It can also improve the roughness of 3D-printed parts [360,361,362]. However, some surface modifications, such as alkali treatment, which is one of the most common surface treatment options, can result in undesirable morphological changes and have an adverse effect on the bulk mechanical properties of PLA constructs [363].

PLA materials have applications other than 3D-printed porous scaffolds, and they show shape-memory effects, which implies that they can switch between a permanent shape and a temporary shape when activated by an external thermal stimulus. When heated above their glass transition temperature, extruded PLA filaments (i.e., 3D-printed) shorten in the printing direction and thicken simultaneously. By rationally placing printed filaments into a multi-layer construct, a complex 3D structure can be obtained after the flat construct is thermally activated [176]. These unique features have been used in the design of 4D-printed objects, including the shape-shifting of flat constructs to pre-programmed 3D shapes [96,97], and reconfigurable [364] and deployable [127,128] mechanical metamaterials, which employ design strategies such as instability-driven pop-up (Figure 3a), self-folding origami (Figure 3b), and sequential shape-shifting (Figure 3c). PLA materials can also be used to form moulds that can later be used to create soft mechanical metamaterials with shape-matching properties [95] (Figure 3d). Furthermore, PLA materials can be used for the design and fabrication of low-cost prosthetics, such as hand prosthesis and artificial fingers (Figure 3e) [365], and non-assembly mechanisms for medical devices [366,367].

#### 4.3.3. Polycaprolactone (PCL)

PCL, a semi-crystalline poly (α-hydroxyester), is a low-cost polyester characterised by its remarkable viscoelastic and rheological properties upon heating. These features make it an excellent candidate for melt-based extrusion printing. It is also a proper thermoplastic material for FDM, owing to its low melting point and high decomposition temperature (350 °C) [370,371]. PCL is also biodegradable, as it resorbs slowly by hydrolysis owing to its high crystallinity and hydrophobic properties [372,373]. Its degradation period is more extended than that of polylactide, making it suitable for applications requiring long degradation times. During the more extended degradation period, the structural stability of the scaffolds can be substantially enhanced, and the rapid degradation of natural polymers can be counterbalanced [374].

The degradation mechanism of PCL is controlled by microorganisms or hydrolysis of ester linkages in a physiological environment [375]. Its nontoxic nature and excellent tissue compatibility make it a popular material for implantable devices. It has been widely utilised in resorbable sutures, biodegradable scaffolds in regenerative medicine, and drug delivery mechanisms [372]. Other PCL applications include scaffolds for tissue engineering of bone and cartilage [376].

However, due to a lack of a bioactive surface and cell adhesion properties, as well as its hydrophobicity, the cell adhesion and proliferation of PCL require improvement [373]. For example, the surface of electrospun PCL nanofibers can be improved by applying various methods, such as plasma treatment, physical adsorption or surface coating of drugs, proteins, and genes, and surface graft polymerisation [377].

Moreover, the introduction of other materials, such as natural polymers in scaffolds, can provide some beneficial properties, including better ductility, biocompatibility, biodegradability, and so forth. When alginate and PCL are mixed via FDM, the composite scaffolds show significantly improved wetting behaviour and water absorption characteristics compared to those of pure PCL scaffolds [378]. Biological properties, such as cell-seeding efficiency, calcium deposition, osteoblast cell viability, and alkaline phosphatase activity, have also been improved due to the alginate constituent [379]. A combination of electrospinning, 3D printing, and a physical punching process is also employed to manufacture PCL/alginate fibrous scaffolds to further enhance the cellular adhesion properties of PCL. The scaffold alginate content vastly improves the hydrophilic properties and water absorption characteristics compared to those of PCL scaffolds, which is beneficial with respect to cell viability, proliferation, and osteogenic differentiation [380].

PCL can also be co-deposited (printed along) with calcium phosphate (CaP), followed by sintering to manufacture a scaffold, or coated on the surface of printed and sintered CaP scaffolds to improve the mechanical strength and elastic modulus of CaP-based materials [381,382] (Figure 4a). CaP has been developed as scaffolds for bone growth and approved as bone fillers by the U.S. Food and Drug Administration (FDA). However, when used alone, it is not capable of providing adequate mechanical properties for hard tissue repair or replacement. In order to establish stable scaffold amalgamation within the host body and ensure that successive regeneration of the host tissue develops continuously, adequate mechanical strength is required. This implies that the compressive strength, elastic strength, tensile strength, and fatigue strength of the polymer/ceramic scaffold must all be sufficient at load-bearing sites and maintained at a sufficient level after implantation until new tissue is ready to restore function [382,383].

Interpenetrating hydrogels with various densities of pectin-g-PCL and gelatine methacrylate resulted in the development of strong hydrogels with enhanced mechanical properties (i.e., compressive and tensile moduli) following double crosslinking by UV light and Ca^2+^ ions, whereas crosslinking only by UV light alone led to a reduction in mechanical properties [372]. These hydrogels were observed to promote the ingrowth of pre-osteoblasts cells in vitro and hence were found to have excellent potential for bone tissue engineering [386].

#### 4.3.4. Poly(lactic-co-glycolic) Acid (PLGA)

PLGA is a biomaterial that has been widely used in the production of drug-releasing devices due to its excellent biocompatibility and controllable biodegradability properties [387,388,389]. PLGA is simple to process, and AM techniques can be applied for scaffold fabrication and bone reconstruction in tissue engineering [390].

There are various methods for controlling the degradation rate, such as altering the molecular weight of the polymer and changing the ratio of its ester linkages of glycolic acid to lactic acid (LA) [391]. A higher percentage of LA results in less hydrophilic PGLA, and hence, the degradation rate is lower because less water can be absorbed by the polymer [392].

Sole PGLA has weak mechanical properties and cell affinity, and it is commonly preferred to be compounded together with a ceramic constituent to form a polymer/ceramic composite scaffold for tissue engineering applications. Consequently, composites with polymer matrices, including biologically active nanoparticles, have gained particular attention in the biomedical field [393,394].

Solvent casting [395], fibre spinning [396], electrospinning [397], and dip coating [398] are some of the methods that can be used to produce medical devices composed of PLGA materials. In addition, 3D printing can also be used to manufacture PLGA devices for drug delivery purposes because of its adaptability in producing optional configurations and the ability to fine-tune the placement of drug-loaded substances [208,399,400]. By using 3D printing fabrication technologies, additional parameters, such as geometry, porosity, and polymer composition, can also be tailored [401,402,403,404,405,406].

PLGA parts can be printed using either low-temperature solvent-based or high-temperature solvent-free processes [399,400]. High-temperature processes are not suitable for heat-sensitive drugs because they require temperatures greater than 95 °C, while the glass transition temperature of PLGA is 35 °C to 60 °C [399,407].

However, harsh solvents are generally used in low-temperature fabrication methods, potentially denaturing the incorporated drugs [408,409]. As the presence of organic solvents in PLGA parts can be harmful to the body during the bioprinting of PLGA, special care must be taken to remove the solvent in the low-temperature fabrication method in the context of drug delivery applications [410].

Different solvents can be used for the 3D printing of PLGA parts. Dimethylacetamide is used as the ink for printing PLGA parts for loading drugs. However, because it has a high boiling temperature, only 2D structures can be created using this solvent [411]. Extrusion-based systems can use solvents, such as chloroform [412], tetraglycol [400], and acetone [413], in order to overcome the rheological limitations of inkjet systems.

Essentially, applying these solvents also results in poor printability or undesirable leaching of the solvent from the scaffolds after printing. The leaching of the solvent is not appropriate for an “ideal” drug delivery process, and it can cause toxicity during in vivo drug release or in the course of in vitro studies [414]. Therefore, mild solvents must be introduced for 3D printing of PLGA to manufacture drug-releasing biodegradable products.

Methyl ethyl ketone (MEK) has been used as a mild organic solvent in a recently developed novel low-temperature 3D printing technique for developing PLGA constructs [414]. MEK has been found to be a promising solvent in the 3D printing of PLGA devices that are parts of drug release systems. MEK application results in printed constructs with high shape fidelity, from which MEK can be removed following the printing procedure [414].

#### 4.3.5. Proprietary Polymers

Proprietary polymers refer to commercial (photo-resist resin) polymers that have been widely used in various (high-precision) 3D printing processes. The chemical compositions of these classes of polymers are often unknown and cannot be altered. Vero^TM^ and Agilus^TM^ are examples of UV-photo-cured polymers used in Stratasys^TM^ Polyjet 3D printing machines. A combination of these polymers with different shore harnesses already exists, which allows for a wide range of elastic stiffness properties in multi-material 3D printing. These materials have been widely used to mimic the bio-inspired design features of biological materials [33,80,81,82]. They have also been used in the design of multi-material mechanical metamaterials [184,415] with programmable properties, such as strain-rate dependency [99] and controlled buckling-driven functionalities (Figure 3f) [98]. Such materials have several applications in soft robotics and exoskeletal devices.

Other examples of such commercial resins are IP-Q^TM^, IP-S^TM^, and IP-L^TM^, which are used for 3D micro-fabrication with high resolution using a nano-scribe^TM^ machine that works on the basis of direct laser melting using two-photon polymerisation (Figure 3g). This fabrication method is used for the surface modification and decoration of biomaterials through the addition of nano-patterns (Figure 3h). The printing process can be adjusted in a way to easily fabricate large areas of nano-topographical features. Nano-topographical features can be printed at the submicron level in the form of nano-pillars. Nano-pillars can act as a mechanical killing mechanism to kill bacteria while keeping cells alive. This has been reportedly achieved by adjusting geometrical parameters, such as interspacing, height, and shape, using computational modelling [416,417]. Different geometrical designs can be fabricated using AM processes at very high resolutions. Furthermore, the physical properties (e.g., wettability) of such surfaces, the mechanical properties [418], and their interactions with human cells can be analysed (Figure 3i) [369]. Topographical features can also be incorporated into microfluidic systems [368]. Electron beam-induced deposition (EBID) is another 3D printing technique that has been used for fabricating objects with features at the nanoscale. It works on the basis of dissociating precursor molecules (i.e., trimethyl-platinum (IV)) using a focussed electron beam. The killing efficiencies of different types of bacteria (e.g., *Escherichia coli* and *Staphylococcus aureus*) in relation to various types of nano-pattern distributions have been analysed [419,420,421]. The mechano-bactericidal effects of such nano-pillars have also been investigated using atomic force microscopy [422].

### 4.4. Composites

Polymer composites or polymer matrix composites are obtained by incorporating reinforcements of particles, fibres, or nanomaterials into polymers. This results in better mechanical properties and functionality. Such composites are extensively used in a wide range of medical applications, including dental treatments, regenerative medicine, and tissue engineering. The materials that are used for these applications must be biocompatible and have the required mechanical and physical properties. A bio-composite is also classified as a composite that contains natural reinforcing fibres [423]. AM of composite structures has attracted a lot of attention recently due to its flexibility and the ability to produce high-performance products while being able to control the geometry of composite structures and constituents and minimising waste [424].

#### 4.4.1. Particle-Reinforced Polymer Composites

Particles can be easily and economically incorporated into the polymer matrix either in powder form or liquid form, depending on the method of 3D printing. They can greatly enhance the physical and mechanical properties of the product; for example, adding iron or copper [425] particles or glass beads can improve the tensile modulus of the polymer matrix [426].

#### 4.4.2. Fibre-Reinforced Polymer Composites

Glass fibres [427] and carbon fibres [428,429] are the most preferred reinforcements used for polymer matrix composites to enhance their mechanical properties. In addition to the type of reinforcement, the orientation and void fraction of the fibres determine the properties of the final printed product [430]. During the 3D printing process, some voids may be formed, which can affect the mechanical properties of the final 3D-printed structure [431]. The porosity of 3D-printed parts due to voids can be significantly reduced by adding expandable microspheres to the polymer [432]. To date, it has been nearly impossible to print continuous fibres, and only short fibres could be 3D printed. Recently, there have been major developments in establishing the relationship between process parameters and printed composite specimens [433,434].

Another major development concerns shape-memory polymer composites that can expand 3D printing to 4D printing technology by incorporating the time factor. These composites are of great interest due to their ability to recover deformation [435]. In addition to temperature-responsive shape-memory composites, water-responsive shape-memory composites have also been developed by applying AM methods [175]. In the cited study, a composite composed of cellulose fibrils and acrylamide changed its shape when immersed in water.

#### 4.4.3. Nanocomposites

Nanomaterials have also been incorporated into matrix materials to enhance their mechanical properties. Shofner et al. [436] demonstrated that the addition of 10 wt% carbon nanofibre could increase the tensile strength of 3D-printed parts by up to 39%, although the elongation decreased, and brittleness increased. In another study, the addition of just 0.2% graphene oxide to an SLA-fabricated photopolymer composite caused increases of 62% in tensile strength and 12.8% in elongation [437], which is quite remarkable. The introduction of nanomaterials, such as carbon nanotubes, can also significantly improve the electrical properties in addition to the mechanical properties of the composite [438].

Functionally graded polymer nanocomposites have been realised by 3D printing. This can be performed by introducing different volume fractions of nanomaterials to specific locations of the structure of the part [439].

## 5. AM of Biomedical Ceramics

According to the definition of Richerson [440], “*most solid materials that aren’t metal, plastic, or derived from plants or animals are ceramics*”. Kingery [441] defined ceramics as non-metallic and inorganic solids, and they can be found in the form of oxides, nitrides, and carbides, which is thus far the most widely accepted definition. Inorganic semiconductors, diamond and graphite, for example, all belong to the category of ceramics. Here, we refer to ceramics as non-metallic and inorganic solids that can have metallic components, and for their formation, they can be subjected to the heating process for hardening purposes.

Ceramics may contain a variety of covalent, ionic, and metallic bonds, distinguishing them from many solid molecular iodine crystals, such as individual I_2_ molecules and paraffin wax composed of long-chain alkane molecules. Ceramics are generally considered to be hard, corrosion-resistant, and brittle materials [441]. Recent advances in ceramics have introduced many new possibilities for practical applications. Advanced ceramics have been favoured as one of the most important materials for various industrial and medical applications in recent years. Investigating, producing, and employing solids with ceramics as the main constituent is a field that is distinguished as ceramic science or industry. This can also include research concerning the refinement of raw materials, the development of new products from chemical mixtures, and the individual characteristics of their components [442].

### 5.1. Classification of Ceramics

Ceramic products have a wide range of applications, ranging from simple building tiles to advanced magnetic components and electronic modules. They can be classified as traditional and advanced ceramic materials. The major developments of advanced ceramics occurred in the 20th century [442].

Ceramics can also be categorised as monolithic ceramics and composite ceramics based on the number of chemical constituents. Monolithic ceramics can be further classified into two categories: crystalline solids and amorphous materials. Crystalline solids can be single-crystal or polycrystal, while amorphous materials can be glass or other amorphous non-crystals, such as amorphous silicon. An example of composite ceramic is concrete [443].

### 5.2. Properties of Ceramics

The electrical and thermal properties of metals are controlled by loose electrons. In ceramics, however, the valence electrons are bound, not loose, resulting in poor thermal and electrical conduction. Exceptions are unavoidable; for instance, diamond, which is also classified as a ceramic, has the highest known thermal conductivity [444].

Ceramics exhibit atypical compressive and tensile properties, which differ from those of metals and polymers, and this is a vital design consideration when using ceramics in load-bearing applications [444]. Moreover, the toughness of ceramics is often relatively low. To compensate for the low toughness, they can be mixed with other materials, such as metals or polymers, to form composites [445].

The most important and common mechanical characteristics of ceramics are linear elastic deformation and brittle fracture under tension [446]. Most ceramics (e.g., polycrystalline alumina) are reasonably elastic at room temperature, and some other ceramics, such as MgO single crystals, show slight residual deformation when the stress is relieved [447]; however, this minor non-linearity is often neglected.

Stochastic strength behaviour is another characteristic of ceramics. This can be seen when testing identical ceramics. This behaviour is related to flaws in the microstructure and the effectiveness degree of the flaws. Such behaviour has to be taken into consideration when ceramic materials are used in the design of any product [448].

Ceramics exhibit time-dependent material properties. Creep, which is defined as time-dependent deformation under constant applied stress, is an example [442].

Permanent deformation behaviour can be easily observed in the viscous behaviour of a liquid. At elevated temperatures, some ceramics, such as glass, also act as extremely viscous fluids, and under these conditions, the consideration of liquid-like behaviour is appropriate. There have been many studies so far to determine changes in viscosity as a function of temperature [449].

Another type of permanent deformation is plastic deformation, which occurs mostly in metals but can also be observed in ceramics at high temperatures [450]. The parameter χ, called the “brittleness measure”, has been introduced to study and define the deformational characteristics and inelasticity of ceramics. In some ceramics, such as aluminium oxide reinforced by zirconium dioxide and zirconium dioxide stabilised by yttrium oxide, χ is equal to 1, which indicates that they obey Hooke’s law. In another category of ceramics, χ is 1, which implies that they hold on to residual stresses. Some examples of the second category are cordierite, silicon nitride with boron nitride, corundum refractory material with zirconium dioxide, and zirconium dioxide stabilised by magnesium oxide [451].

### 5.3. Manufacturing Methods for Ceramics

Ceramics can normally withstand harsh operating environments, which can be beneficial to many industries. Owing to a wide range of favourable mechanical properties and characteristics, chemical inertness, and excellent features at high temperatures, ceramics are desirable materials in biomedical applications. However, advanced manufacturing techniques are required to achieve superior properties as well as efficient production [390].

Ceramic parts can be produced by using a variety of existing methods, and they can be classified into the following five categories [452]:Casting/solidification methods: in this category, the liquid and solid states of the starting material change, and this is accompanied by some volumetric changes in most cases.Deformation methods: in this category, ceramic structures are formed through a plastic deformation process.Machining and material removal methods: an abrasive process is applied to remove the material from a ceramic block.Joining methods: in this category of methods, different ceramic bits and pieces are combined using various joining techniques.Solid free-form fabrication methods: this category of methods includes various AM methods for the fabrication of ceramics.

The first four categories are considered to be among the conventional methods of fabricating ceramic parts. In the last category, several AM techniques are suitable, and nearly all of them, except for material jetting, have been used to produce ceramic structures. To achieve complex bulk or porous materials, AM technologies, such as SLS, lithography-based ceramic manufacturing (LCM), SLM, and FDM, have been adopted.

Geometry design is a significant factor in ceramic AM, and it has received considerable attention in recent studies [453]. The operating conditions of AM ceramic products dictate some other required characteristics, e.g., certain electrical and mechanical characteristics. The use of ceramics, in combination with other groups of materials that have different properties and are realised by applying suitable co-manufacturing procedures, is becoming more extensive than ever before and, in some cases, even essential for the development of novel biomaterials and medical devices [385].

To meet the requirements for certain applications, direct AM has been proposed and widely used to produce ceramic parts and ceramic-reinforced metal matrix composites. A high-power-density laser beam is used to generate heat in the AM process. In several ways, the laser deposition-additive manufacturing (LD-AM) technique outperforms other direct AM methods with respect to production performance, the ability to remanufacture components, and the production of functionally graded composite materials; however, issues such as poor bonding, cracking, and lowered toughness persist in LD-AM-built products [454]. However, in practice, a binder, usually a polymer with a low melting point, is used in indirect AM to help consolidate the layers during AM. The binder is, in most cases, removed through a process called debinding, followed by sintering.

Although AM has been more successful in metals and polymers than in ceramics, there is a growing interest in using AM technologies to produce high-quality dense ceramic products. The appropriate technique is determined by the sizes, shapes, binder concentrations, and surface conditions of the proposed product, as well as the type of ceramic used [455].

#### 5.3.1. Powder-Based 3D Printing (P-3DP)

In regard to ceramic materials, binder jetting and SLS are the two common and globally accepted powder-based 3D printing (P-3DP) processes [456].

##### Binder Jetting

There are two main categories of material-binder arrangements that can be employed for producing scaffolds: (*i*) the ceramic powder is mixed with an organic binder that can be dissolved in water or a solvent and sprayed on the printing bed [454], or the ceramic powder can be combined with a polymer that can act as a binder [455,457]; (*ii*) a reactive liquid binder that allows low-temperature activation and promotes the densification of ceramic powder at relatively low temperatures is applied. Many studies have been conducted on calcium phosphate, in which the binder is phosphoric acid [458].

For some biomedical applications, polymer-derived ceramics have been developed by means of P-3DP of a preceramic polymer powder containing no inert or active fillers. Without the use of external binders, the sprayed solvent can melt the preceramic polymer powder and combine the particles while filling the gaps between them [459]. Following heat treatment, the resultant product can retain some residual porosity, as the product does not experience sintering. To resolve this issue, the preceramic polymer is combined with a glass powder and reactive fillers to produce a bio-ceramic scaffold based on wollastonite–apatite [460]. Reviews on the developed strategies for the material and design of ceramic scaffolds used with the P-3DP method can be found in a number of studies [461,462].

Most of the research in the area of 3D printing of porous ceramic structures based on powders is focussed on producing scaffolds for tissue engineering applications. The pores in the scaffolds must be in a range of 50 to 1000 μm, and the scaffolds must contain a porosity greater than 60%, which is required for productive bone ingrowth and proper vascularisation of implants. Another important dimension is the residual micro-porosity in the resultant product, which is preferably kept under 10 μm to increase the surface area, which in turn results in better protein absorption and ion transfer. It is important to note that pores smaller than 500 μm cannot be directly printed owing to the limitation on the resolution and difficulty in removing excess powder particles [385].

P-3DP is a suitable method for building porous ceramic structures. An important restriction in using P-3DP is the low density that can be achieved, partly due to the low powder-packing density in the powder bed, thereby limiting the design variation and automatic part production. Commonly, the ceramic powder particle size exceeds 20 µm, which does not provide sufficiently high sintering activity for the production of dense ceramics. One of the solutions is to use ceramic slurry instead of the dry powder. In some cases, heat treatment following the manufacturing of the part enhances the density [385].

##### Selective Laser Sintering (SLS)

In the SLS method, the density of the material at each location can be specified by either direct sintering of ceramic powder or by combining it with a binder, such as a polymer or an inorganic material that is melted by using a laser beam. Most ceramics are rather stable when exposed to high temperatures, and therefore, direct sintering is not a straightforward method for their fabrication. Moreover, the limited duration of the laser action on the powder results in inappropriate sintering [463,464,465,466] owing to a lack of extensive atomic diffusion and thus insufficient neck formation and growth.

Incorporating thermally activated binders into ceramic powders has resulted in acceptable porous ceramic structures [467,468]. Another application in this regard is selected laser curing, developed by Friedel et al. [469], in which preceramic powder is used to fabricate a polymer-derived ceramic part.

Kolan et al. [470] manufactured bioactive glass scaffolds with a porosity of 50% and pore sizes between 300 and 800 μm by applying a polymer binder, SLS, and subsequent debinding and sintering at 675–695°. SLS is also used to fabricate hydroxyapatite–silica scaffolds for bone replacement with pore sizes of 750 to 1050 μm and porosities of 25% to 32%. For this purpose, a slurry composed of hydroxyapatite powder and silica sol as a binder is used in SLS, and the subsequent sintering is performed at 1200 °C [471].

Another possibility in the development of scaffolds for tissue engineering purposes is applying biocompatible polymers as binder, for example, a scaffold developed from poly (l-lactide-co-glycolide)–hydroxyapatite (HAP) and β-tricalcium phosphate (β-TCP) or from polyetheretherketone–hydroxyapatite as bone substitutes [472,473]. The manufactured scaffolds are actually biopolymer–ceramic composites, and no further post-processing is needed [385].

SLM is another option being explored, as the presence of a liquid phase ensures rapid densification [474]. Mixtures of alumina and zirconia powders with sizes of 20 and 70 µm are used as feedstock materials to achieve a high packing density of particles in the powder bed. To minimise the risk of thermally induced stresses in the workpiece, the powder bed is preheated up to 1600 °C prior to printing. The formation of the alumina–zirconia eutectic can decrease the melting points of the individual ceramics (especially ZrO_2_: 2710 °C) to 1860 °C. The above considerations allow the creation of dense ceramic products that can potentially display exceptional mechanical properties. However, there is always the possibility of crack development in ceramic parts, and unrestrained fluid infiltration at the laser point may cause the formation of an undesirable surface on the outside of the workpiece [385].

#### 5.3.2. Stereolithography (SLA)

This approach is based on the photopolymerisation of a liquid resin containing ceramic powder, which is performed in consecutive layers in the same way as other indirect AM techniques. The slurry is composed of a photo-initiator, a monomer solution, and other additives that help in the dispersion of the ceramic powder. The desired volume fraction of the ceramic powder is normally between 40 and 60% [475,476]. The manufacture of dense ceramic products is possible by applying the SLA technique, followed by sintering as a post-processing method [475].

Many investigators have used the SLA method to create high-quality porous ceramic products for a variety of industries. Kirihara [477], for example, used this technique to fabricate ceramic dendrite structures with geometrically ordered lattices and demonstrated that with an acrylic resin, hydroxyapatite scaffolds for tissue engineering purposes could be manufactured with a porosity of 75% and lattice density of about 98% after post-SLA dewaxing at 600 °C for 2 h and sintering at 1250 °C for 2 h were applied. Chu et al. [478] created porous hydroxyapatite structures with a target porosity of 40% and controllable pore geometry of either radial or orthogonal channels from HAP suspension in acrylates.

Bian et al. [479] manufactured a biphasic biomimetic osteochondral scaffold with a bone phase, a cartilage phase, and a transitional structure between bone and cartilage. The scaffold was initially produced by creating a porous β-TCP scaffold using SLA of ceramic suspensions, followed by drying and sintering, and finally, gel casting and freeze-drying of a collagen solution were performed to introduce the cartilage phase. The pore sizes of the bone phase were measured to be 700 to 900 µm, with a porosity of 50 to 60 %, while those of the cartilage phase were 200 to 500 µm.

#### 5.3.3. Extrusion-Based 3D Printing: Robocasting, Direct Ink Writing (DIW), and FDM

One of the most popular AM methods for the fabrication of porous ceramic structures is the direct deposition of slurry. In this process, viscous ceramic paste is extruded through a nozzle in the shape of a filament, and then it undergoes a transformation from pseudoplastic to dilatant by extruding and drying in air. Air drying restricts the minimal calibre of the nozzle to 500 µm in order to avoid clogging. A solution to this issue is the development of special inks with reversible gel conversion, which is known as robocasting (Figure 4b) [480]. The ink initially acts like a viscous gel in the printing head, but the shear stress of the extrusion disrupts the internal structure of the gel and significantly reduces the viscosity. The viscosity expands again following extrusion. The rheological characteristics of the filament must be properly arranged to prevent its distortion and bowing, particularly in cases where there are spanning features in the configuration of the part [385].

Many studies have been conducted to explore the use of the robocasting process for the fabrication of porous bio-ceramic scaffolds. Genet et al. [481] investigated the mechanical behaviour of robocasts and sintered porous hydroxyapatite scaffolds based on the Weibull theory. From the experimental data, they demonstrated that the expansion of porosity resulted in a reduction in compressive strength. By using a special ink with a thermally reversible gel, Franco et al. were able to create scaffolds out of HAP, tricalcium phosphate (TCP), and biphasic calcium phosphate [482]. They also realised that by increasing the gel content, the micro-porosity of struts could be substantially increased from 5 to 40%, resulting in a reduction in bending strength from 25 to 2 MPa [482].

Miranda et al. [483] succeeded in creating HAP scaffolds with an overall porosity of 39% and a strut microporosity of 5% and investigated their failure modes under uniaxial loading conditions. They demonstrated a value of approximately 50 MPa for the compressive strength of the fabricated scaffolds and found that the strength could be increased up to twofold by placing them in a simulated body fluid [384]. Fu et al. [484] created relatively strong bioglass scaffolds with dense rods of 100 μm in diameter and an overall porosity of 60% with unidirectional pores. The mechanical strength in the direction parallel to the pore channels reached 136 MPa, while the strength in the direction normal to the pore channels was 55 MPa.

Porogens can be added to ceramic paste to obtain porosity at different levels [485]. Dellinger et al. affixed poly(methyl methacrylate) (PMMA) particles to the ink and built scaffolds with three ranges of porosity. Macro-pores in a range of 100 to 600 μm were created by arranging and locating them among rods of HAP; micro-pores with sizes of 1 to 30 μm were generated within the rods by introducing PMMA particles; and sub-micro-pores with sizes under 1 μm were obtained as the output of imperfect sintering [485].

FDM of ceramics has been introduced as one of the methods that involve the extrusion of ceramic paste and is similar to the methods used in the manufacturing of polymer parts. In this method, a mixture of ceramic and polymer powder that liquefies during extrusion is extruded and returns to the solid phase as it cools down. Following 3D printing, the polymeric part is removed, and the remaining ceramic part is sintered. Grida et al. [486] used a combination of 55 vol% zirconia with wax and extruded the feedstock by applying nozzles with calibres from 76 to 510 μm. Park et al. [487] used a paste containing 40 wt% HAP mixed with molten PCL at a temperature of 120 °C and extruded the feedstock with a nozzle size of 400 μm and a scaffold strand distance of 600 μm. Kalita et al. [488] also used FDM to fabricate polypropylene–TCP composite scaffolds with a compression strength of 12.7 MPa, an overall porosity of 36%, and an average pore size of 160 μm.

In general, scaffolds constructed with DIW techniques are mechanically stronger than those built with powder-based methods [385]. However, in the case of powder-based indirect AM techniques, there are fewer geometry restrictions, and therefore, constructing cylindrical shapes with either radial or orthogonal pores is preferred because the resulting structure is more similar to natural bone than that obtained with DIW methods [385].

The scaffolds produced by P-3DP methods have sintering necks between the original powder particles, resulting in a high residual porosity [489], while the ones produced by DIW can be subjected to sintering to become more dense (Figure 4c). However, absolutely dense scaffolds cannot be constructed by this method owing to the unavoidable presence of approximately 15% micro-porosity in the scaffold [483]. The remaining micro-porosity is useful for tissue engineering, but it may have an adverse effect on the mechanical properties of the struts [385].

#### 5.3.4. Negative AM Techniques

By applying negative replica methods, some restrictions concerning the shape and functioning of the products can be overcome. In these techniques, AM is used to create a polymer mould, which can then be filled with ceramic slurry. Subsequently, the polymer must be dissolved, followed by ceramic sintering to produce the finished product [490].

The initial polymeric mould can be created using any of the AM methods. Detsch et al. printed a wax mould and filled it with HAP slurry in their study. They also duplicated the exact shape using the robocasting method and obtained 44% porosity, whereas negative AM produced 37% porosity. The scaffolds created using both methods had the same pore and strut thickness dimensions [491]. The SLA method can also be employed to create resin moulds for HAP scaffolds with 50% porosity [492]. In the study conducted by Woesz et al., a gel-casting method was applied to enhance the mechanical properties of the ceramic green body [492].

Freeze foaming is another method for producing porous ceramic structures. The most common foaming processes are generally those originating from the exhaustion of environmentally harmful unstable organic pore-formers or whole polymer scaffolds [493].

When the freeze foam materials are composed of HAP or zirconia (ZrO_2_) or their composite mixtures, biocompatible or bio-inert products can be manufactured based on the properties of the individual materials [494,495].

By combining the LCM and novel freeze-foaming methods, biocompatible structures can be obtained, which may be the next generation of bio-composites. They can result in a combination of dense and porous structures in an isolated product. Although AM technology has the advantage of allowing for customised features, freeze foaming creates porous structures with adaptive pores and porosities, allowing for the growth and differentiation of mesenchymal stem cells [493].

### 5.4. Biomedical Applications of Ceramics

With the advancement of ceramic technology, state-of-the-art ceramics have been developed for biomedical purposes [496]. The first scientifically managed medical uses of ceramics were in dentistry, where porcelain is used to make crowns, and in orthopaedics, where plaster of Paris gypsum (calcium sulphate dehydrate) is used to treat fractures [497].

Today, most research and development on dental prosthetic restorations are focussed on ceramics rather than metals, as ceramics have an advantage because of the white to ivory colour of their oxides [401]. In fact, in addition to mechanical properties, aesthetic considerations, such as colour and translucency, are prioritised in dental applications. In tooth repairs without the application of metals, the colour of the soft tissue maintains a higher resemblance to that of the native tissue compared to porcelain combined with metallic elements. Furthermore, ceramics are not susceptible to corrosion or galvanic effects that are inevitable in the case of metals [401].

Yttrium-stabilised tetragonal zirconia has recently been recognised as another appropriate choice for many dental applications, but observations of in vitro stability demonstrate that aging can be an issue [498]. There are not many studies available regarding its prolonged in vivo durability in oral environments [498].

Bio-ceramics can be classified into two groups, namely, bio-inert and bioactive [498]. Unlike bio-inert ceramics, bioactive ceramics must provide adequate surface conditions for cell adhesion and bone growth [499]. The most common bioactive ceramics contain calcium phosphate components, such as HAP and TCP, which are similar to the mineral constituents of bone to a great extent [499].

Bioactive ceramics have been mainly utilised as coatings on metallic orthopaedic implants. This is especially important in areas where a robust interface with the bone is needed, such as femoral stems and metal-backed acetabular cups in hip prostheses or tibial and femoral stems in total knee replacement systems [498].

Owing to the osteoconductive properties of calcium phosphates, they have been widely used as artificial bone grafts and substitutes for autografts and allografts since the mid-1980s [499]. Unlike natural grafts, artificial bone replacements do not require invasive surgery and can be supplied in large quantities [500]. Another advantage, in comparison to allografts, is a lower risk of rejection and disease transmission. The most common bone replacements are porous structures constructed from biphasic calcium phosphates, such as HAP-TCP composites [500].

The rate of absorption of TCP is higher than that of HAP, allowing for the management of the gross degradation rate of the HAP-TCP composite and customisation of the composite material for the patient, for example, faster resorption for patients with more rapid bone reconstruction [500].

However, recent bone replacements based on calcium phosphate have not been completely effective because porosities at the micro- and nano-scales can cause a variety of physical and chemical features that can affect biological characteristics [500].

Another shortcoming of ceramics is their brittleness and low crack resistance [501]. As inherent brittleness is a critical weakness of ceramics, there are a large number of review articles on the failure of ceramics [502].

## 6. Conclusions and Future Research Directions

Major biomedical applications of biomaterials can be summarised as scaffolds for bone repair, tissue regeneration, reconstructive and orthopaedic implants, cardiovascular devices and prostheses, dental restorations, ophthalmic devices, and drug delivery systems. These products are manufactured from a wide range of biocompatible materials, including metals, polymers, and ceramics, or a combination of these materials.

AM technologies have provided excellent opportunities for the ease of manufacturing and cost-effective production of many new and advanced products from various materials for the abovementioned biomedical applications.

All of the AM techniques in the ASTM classification, especially material extrusion (e.g., FDM), directed energy deposition (DED), material jetting (e.g., Polyjet), PBF (e.g., SLS, SLM, DMLS, and EBM), and binder jetting, are not equally developed and used for medical devices and biomaterial fabrication. The capabilities, limitations, pros, and cons of each technique and associated materials (e.g., metals and their alloys, polymers, and ceramics) as well as considerations for the AM fabrication of biomaterials such as printing speed, part sizes, degree of anisotropy, achievable resolution, the possibility of embedding cells in feedstock materials, the need for support, the need for post-processing, and printing costs, all are important factors that need to be taken into account. The success of each of these 3D printing processes relies, to a large extent, on the employment of optimised or suitable process parameters within the capabilities of the available AM machines.

Aside from selecting the proper AM techniques and suitable printing parameters, the microarchitecture design of biomaterials is one of the critical aspects of their development. It is often necessary to design porous or lattice structures for biomedical applications. This implies that pores with certain morphologies and sizes inside the biomaterials must be fully open and interconnected to allow for the transport of nutrients and oxygen to cells.

In addition to new horizons in producing biomedical devices and products, the versatility of AM methods in enhancing the properties of materials opens up the possibilities of new breakthroughs in the biomedical engineering industry. The possibility of the customisation of design as well as properties is a major consideration in the research and development of AM.

Considering the wide range of requirements in tissue engineering and artificial organs, the current developments of biomaterials seem to be far from satisfactory and require more research in the future. The prolonged existence of biomaterials in the body without immune rejection is still an issue in most cases, which needs more research. Better materials are expected to satisfy the requirements of artificial joints with respect to wear reduction and durability. In addition to biocompatibility, cellular responses to the biomaterial are expected to acclimate with the host tissue in most tissue engineering cases. This necessitates the addition of some bioactive factors to stimulate the desired responses or to prevent a specific reaction, which can be a future path for the development of biomaterials in the AM industry. Another future direction of research in regard to AM is related to micro- and nano-printing of multi-materials, especially bimetals, and the joining methods of various metals, as well as multi-material 3D printing. It is expected that by using AM techniques, in many cases, the issues of conventional joining methods, such as welding and soldering, can be overcome.

## Figures and Tables

**Figure 1 materials-15-05457-f001:**
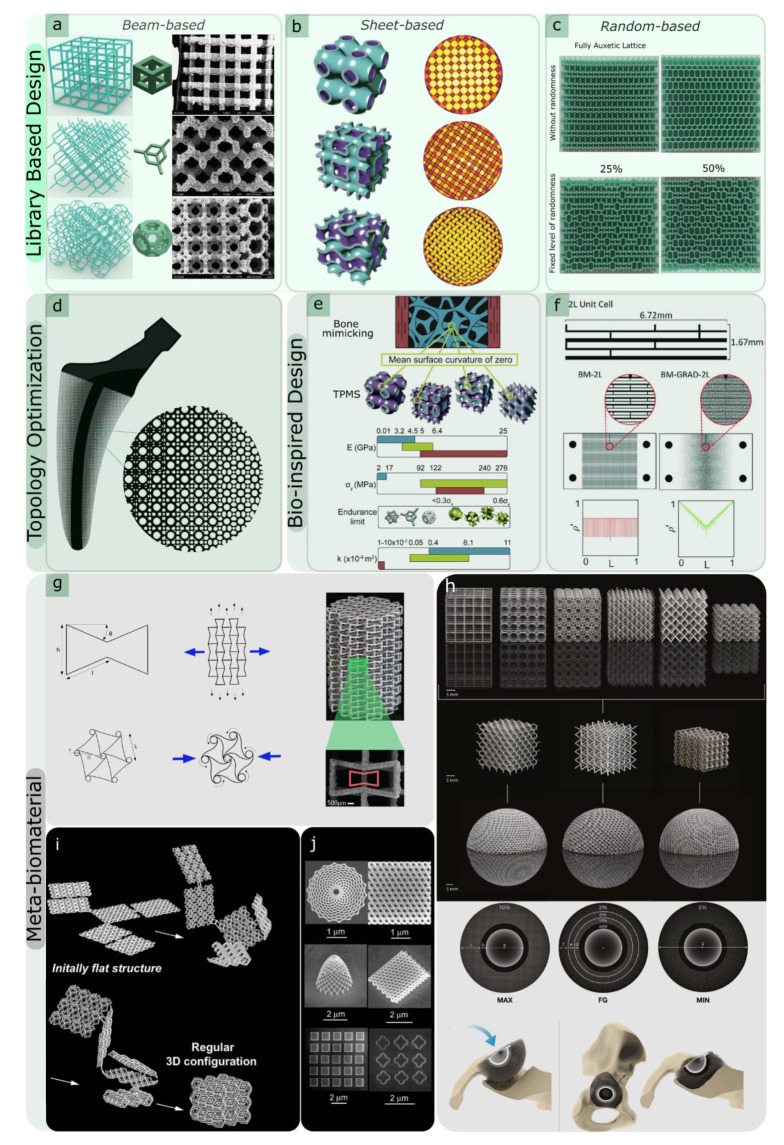
(**a**–**c**) Library-based designs: (**a**) beam-based unit cells, such as cubic, diamond, and truncated cuboctahedron (reprinted from Refs. [26,29] with permission, Copyright 2022 Elsevier), (**b**) surface-based unit cells, such as triply periodic minimal surfaces (TPMS) (reprinted from Ref. [30] with permission, Copyright 2022 Elsevier), and (**c**) disordered and random-based network structures (reprinted from [31] with permission, Copyright 2022AIP Publishing); (**d**) topology optimisation employed in an orthopaedic implant (reprinted from Ref. [32] with permission, Copyright 2022 Elsevier); (**e**,**f**) bio-inspired designs, such as functionally graded hierarchical soft–hard composites inspired by (**e**) bone (reprinted from Ref. [30] with permission, Copyright 2022 Elsevier) and (**f**) nacre-like design exhibiting brick-and-mortar hierarchical unit cell structures (reprinted from Ref. [33] with permission, Copyright 2022 John Wiley and Sons); (**g**–**i**) meta-biomaterial designs: (**g**) auxetic properties, including re-entrant unit cells and chiral structures [34,35] (reproduced from [34] with permission from the Royal Society of Chemistry), (**h**) non-auxetic unit cells, such as cube, truncated cube, truncated cuboctahedron, diamond, body-centred cubic, and rhombic dodecahedron; three non-auxetic unit cells (diamond, body-centred cubic, and rhombic dodecahedron) were chosen for further evaluation in deformable meta-implants after they were evaluated for their quasi-static mechanical properties [36], (**i**) self-folding of origami lattices [37]; (**j**) 2D and free-form 3D nano-patterns on the surface of flat origami sheets using electron beam-induced deposition (EBID) [37].

**Figure 2 materials-15-05457-f002:**
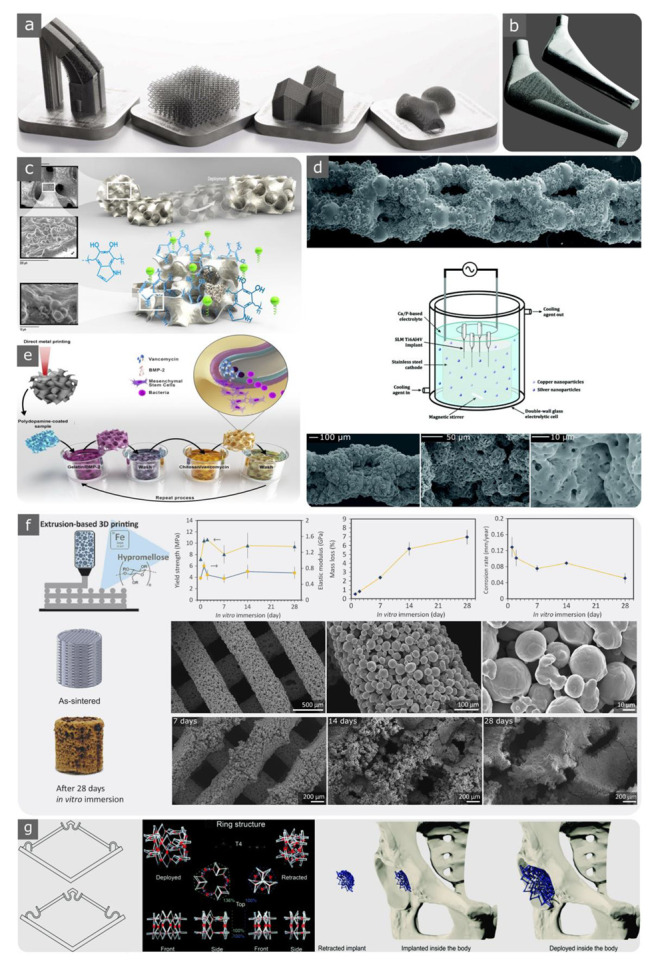
(**a**) Examples of porous metallic structures and bio-implants with various functionalities (reprinted from Ref. [92] with permission, Copyright 2022 Elsevier); (**b**) a hybrid implant that combines solid and porous parts in a single device (reproduced from Ref. [25] with permission from the Royal Society of Chemistry); (**c**–**e**) biofunctionalisation of AM products; (**c**) surface biofunctionalisation of a porous Nitinol structure using polydopamine-immobilised rhBMP-2 (reprinted with permission from [137], Copyright 2022 American Chemical society); (**d**) self-defending additively manufactured implants bearing silver and copper nanoparticles; (**top**) scanning electron microscope (SEM) imaging was used to image the surface morphology of a selective laser melted Ti–6Al–4V implant, (**middle**) a schematic drawing of the electrolytic employed for plasma electrolytic oxidation (PEO) bio-functionalization process, and (**bottom**) SEM images showing the surface morphology after PEO biofunctionalisation at different magnifications (reproduced from Ref. [138] with permission from the Royal Society of Chemistry); (**e**) the layer-by-layer coating process for the biofunctionalisation of additively manufactured meta-biomaterials [139]; (**f**) a schematic of extrusion-based 3D printing process for the fabrication of porous scaffolds; SEM images showing as-sintered and as-degraded iron scaffolds as well as in vitro corrosion products after 7, 14, and 28 days of immersion and the yield strengths, elastic moduli, mass loss percentages, and corrosion rates of the scaffolds before and after in vitro immersion for up to 28 days [145]; (**g**) the principle of deployable implants demonstrated schematically by arranging bi-stable implants (reproduced from Ref. [128] with permission from the Royal Society of Chemistry).

**Figure 3 materials-15-05457-f003:**
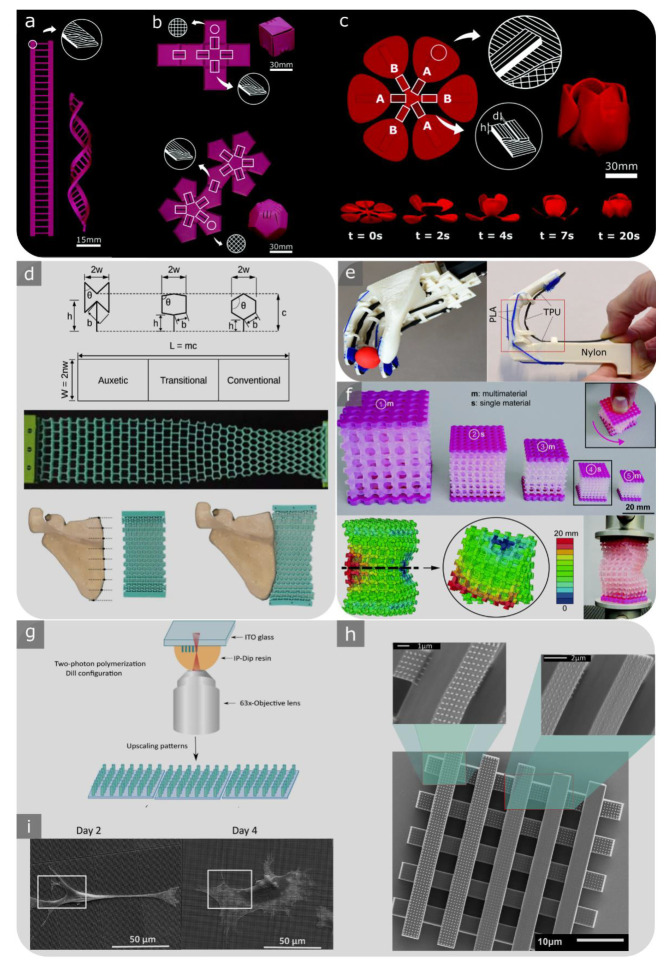
(**a**–**c**) Shape-shifting of the shape-memory polymers, i.e., (**a**) self-twisting: after activation, two flat self-twisting strands form a DNA-inspired shape, (**b**) self-bending: on activation, a flat printed construct is folded into a cubic box, and (**c**) sequential shape-shifting: folding the initially flat petals into a tulip in two steps by controlling the printing directions at specific locations (i.e., A, and B); the time lapses show the folding sequence for both designs (reproduced from Ref. [176] with permission from the Royal Society of Chemistry); (**d**) shape matching of the scapula with a specimen fabricated by three zones of auxetic, transition, and conventional unit cells [95]; (**e**) 3D-printed hand prosthesis [365]; (**f**) buckling-driven soft mechanical metamaterials for external prosthetics and wearable soft robotics, such as exoskeletons and exosuits (reproduced from Ref. [98] with permission from the Royal Society of Chemistry); (**g**–**i**) cell culture using submicron patterns: (**g**) a schematic view of the two-photon polymerisation method [368]; (**h**) SEM image showing submicron-scale topographies incorporated into a porous micro-scaffold [368] with (**i**) cells cultured on patterned surfaces after 2 and 4 days of cell culture [369].

**Figure 4 materials-15-05457-f004:**
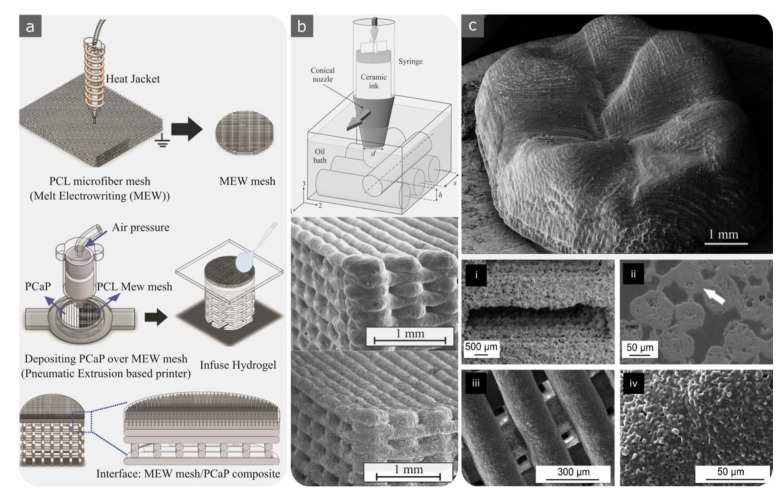
(**a**) An example of a hybrid 3D printing technique used for the fabrication of ceramic-hydrogel connections representing osteochondral interfaces [382]; (**b**) the robocasting fabrication process and SEM images of the scaffolds created by robocasting; the ceramic ink is moved through conical deposition nozzles, which are plunged in an oil bath to create a self-supporting 3D ceramic rod network (reprinted from Ref. [384] with permission, Copyright 2022 John Wiley and Sons); (**c**) SEM micrograph showing the occlusal surface of a zirconia molar crown using the direct inkjet printing technique and SEM images showing hydroxyapatite scaffolds produced by: powder-based 3D printing in (**i**,**ii**); direct ink writing in (**iii**,**iv**) (reprinted from Ref. [385] with permission, Copyright 2022 John Wiley and Sons).

**Table 1 materials-15-05457-t001:** Summary of the different AM techniques, useable materials, their pros and cons, and their biomedical applications.

		Techniques and Materials	Pros	Cons	Biomedical Application
Material Deposition 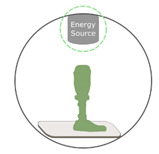	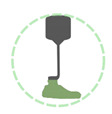	*Material Extrusion* (*FDM*) Hydrogels Thermoplastics Ceramics Bio-inks	+ Low cost + Accessible + Composite materials + Open-source design	- Slow - Anisotropy in printed part - Low resolution - Nozzles impart high shear forces on cells	Bioprinting of scaffolds for cell culture Tissue and organ development Production of rigid and soft anatomical models for surgical planning
	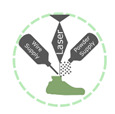	*Directed Energy Deposition* (*DED*) Metal	+ Fast + Composite materials + Dense part	- Expensive - Low resolution - Requires post-processing/machining	Limited use in biomedical application
	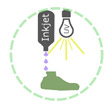	*Material Jetting* (*Polyjet*) Photopolymer Bio-inks	+ Good resolution + Good cell viability + Multiple cell/material deposition	- Slow - Material waste - Limited material selection - Limited fabrication size	Bioprinting of scaffolds for cell culture tissue and organ development (soft tissue)
Powder-based 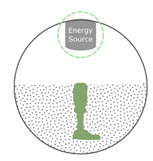	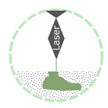	*PBF* (*SLS, SLM, DMLS, EBM*) Thermoplastics Metal powders Ceramic powders	+ High strength and dense parts + Fast + No solvents required + No support required	- Most expensive - Post-processing required	Metallic implants Dental craniofacial and orthopaedic Temporary and degradable rigid implants
	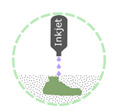	*Binder Jetting* Metal Polymer Ceramics	+ Low cost + Fast + Multi-colour printing + No support needed + Large objects	- Low strength - Requires post-curing and post-processing - Powder poses a respiratory hazard	Degradable metallic implants Generally used for hard, mineralised tissues
Liquid-based 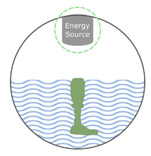	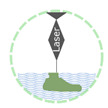	*SLA* Photopolymer Bio-resin Ceramic resins	+ High resolution + Fast + Good cell viability + Nozzle free	- Raw material toxicity - Limited material selection - Possible harm to DNA by UV	Bioprinting of scaffolds for cell culture Tissue and organ development can be used for both soft and hard tissues
	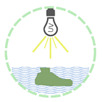	*DLP* Photopolymer Bio-resin Ceramic resins			

**Table 2 materials-15-05457-t002:** Summary of the different approaches for the geometrical design of lattices.

Design Strategy	Method	Geometry/Mechanism Example	Unique Feature	Caution in 3D Printability
**Library-based**	Ordered unit cells	Beam-based: FCC, BCC, octet-truss, and diamond Sheet-based: TPMS, gyroid, diamond, and primitive	Use of (non-)commercial CAD tools Simplicity in geometrical design Originate from crystalline structures Interconnectivity of pores Control of the level of connectivity using either stretching- or bending-dominated unit cells (beam-based unit cells) Control of the localised curvature using sheet-based designs (surface-based unit cell designs)	Design of self-overhanging structure and sacrificial support Limitation in minimum feature sizes (e.g., strut thickness) Orientation with respect to the build plate
	Disordered unit cells	Functionally graded Control of the level of connectivity	Broader range of morphological and mechanical properties Less sensitivity to local defects Straightforward design and fewer complications with overall structural integrity Smooth stress transition using localised geometrical adjustment Independent tailoring of mechanical properties Similarity to biological materials (e.g., bone)	Design of self-supporting struts and their orientations with respect to the build plate Limitation in minimum feature sizes (e.g., strut thickness and orientations)
**Topology optimisation**	Analytical mathematical models and computational approaches to design and obtain optimised microstructures	ESO—evolutionary structural optimisation SIMP—solid isotropic material with penalisation BESO—bi-directional evolutionary structural optimisation	Use of commercial tools and free codes Local microstructural compatibility Creating topology-optimised lattice structures with atypical properties considering multiple objective functions (e.g., negative thermal expansion) Design for multi-functional or mutually exclusive properties (e.g., high elastic stiffness and permeability) Used for tissue adaptation purposes and design of orthopaedic implants	Limitation in manufacturability due to the complexity of the final product Optimisation of the disposition of support materials during AM process to alleviate stress concentrations Acceleration of support removal process
**Bio-inspired design**	Bio-inspired designs	Functional gradient and hierarchical structures	Vast design library of natural cellular materials Multi-functionality and exceptional mechanical properties, such as graded stiffness, using co-continuous multi-material cellular structures Smooth transitions of target parameters in three dimensions and minimised stress concentrations at interfaces	Limitation in minimum feature sizes Use of multi-material 3D printing technology with extreme mechanical property mismatches
	Image-based	Original tissue obtained from non-destructive imaging (e.g., MRI or CT)	Mimicking the functionality and microstructural complexity of the native tissue Creating patient-specific implants and medical devices	
**Meta-biomaterials**	Designer material or mechanical metamaterial	Negative Poisson’s ratio or auxetic behaviour (e.g., re-entrant, chiral, and rotating (semi-)rigid unit cells Non-auxetic (e.g., TPMS-based porous structures)	Unprecedented multi-physics properties (e.g., balance between mechanical properties and mass transport) Tailor-made (mechanical) properties and functionality (e.g., 2D to 3D shape morphing using origami-folding techniques) Stronger interface between the designed part and host tissue Outstanding quasi-static and fatigue performance	Simple to very complex unit cell designs Integration of different unit cells, particularly for the hybrid design of meta-biomaterials
	Kinematic or compliant mechanism-based designs	Multi-stability Self-folding Kinematic mechanisms	Fabricating non-assembly mechanisms with compliant or rigid joints (e.g., metallic clay)	

## Data Availability

Data sharing not applicable.

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
