# Peer review of "Additive Manufacturing of Biomaterials—Design Principles and Their Implementation"

_materials, 2022, doi:10.3390/ma15155457_

Round 1

Reviewer 1 Report

The article is a fairly detailed overview of the current state of the problem of using additive technologies. I want to pay special attention to the volume of articles and a large number of materials considered. On the one hand, this is good, but it can confuse the reader. However, the design of the article chosen by the authors is acceptable and worthy of a positive assessment.

I want to draw the authors to a few points:

The title of the work must contain the word REVIEW

In the introduction, it is necessary to more convincingly state the relevance and purpose of this article.

It is necessary to pay more attention to the methods of obtaining metal powder for additive technologies - to describe in detail the advantages and disadvantages of different methods

3.2. Shape memory alloys: Please expand the section. Properties after printing can be controlled by Thermal Cycling and Heat Treatments (see 10.1007/s40830-022-00363-4)

3.1. Biodegradable metals: Please expand the section. write details about the promising Fe-Mn-Si alloys

The conclusion is too general, it needs to be rewritten

The article contains a lot of drawings from other published works - this is good, but you need to find out if it is possible to reproduce them from the copyright holder

Author Response

Responses to the reviewers’ comments

The authors would like to thank the reviewers for the constructive comments that helped us improve the manuscript. We hope that the revised manuscript together with the replies below will satisfy the expectations of the respected reviewers.

Reviewer #1:

The article is a fairly detailed overview of the current state of the problem of using additive technologies. I want to pay special attention to the volume of articles and a large number of materials considered. On the one hand, this is good, but it can confuse the reader. However, the design of the article chosen by the authors is acceptable and worthy of a positive assessment.

I want to draw the authors to a few points:

We thank the reviewer for their positive feedback. We have included reviewers’ suggestion in the revised version of the manuscript, and we believe this will prevent the above-mentioned confusions for the readers.

Remark 1.1. The title of the work must contain the word REVIEW

The word “Review” has been added before the title to specify the type of the article.

Remark 1.2. In the introduction, it is necessary to more convincingly state the relevance and purpose of this article.

The main objective has been added at the end of the Introduction section:

“...the main objective of this review article is to present a clear picture of how this technology can be applied for producing bio-materials of novel designs, what are the challenges and limitations and where the technology is heading… It is indented to stimulate the further development and widespread application of the technology to turn design ideas into implants and other medical devices as well as those in tissue engineering applications.”

Remark 1.3. It is necessary to pay more attention to the methods of obtaining metal powder for additive technologies - to describe in detail the advantages and disadvantages of different methods

We fully agree with the reviewer that metal powders with a spherical morphology and particle size distributions suitable for many of AM processes are of critical importance and may even be a limiting factor for printing implants and other medical devices of desired materials and designs with desired mechanical, chemical and biological properties at affordable costs. The methods used for fabricating such powders are mostly conventional inert-gas atomisation or contrigual atomisation. There have been many good review articles and book chapters concerning the powder fabrication methods, such as: https://api.intechopen.com/chapter/pdf-download/50676.pdf. The present review article is focused on the design principles, relevant 3D printing processes and applicable biomaterials. Adding powder fabrication methods would dilute the main focus.

Remark 1.4. 3.2. Shape memory alloys: Please expand the section. Properties after printing can be controlled by Thermal Cycling and Heat Treatments (see 10.1007/s40830-022-00363-4)

The following description about a remarkable development in SMAs has been added to subsection 3.2:

“According to the findings of Tsaturyants et al. [176], a combination of thermal cycling and heat treatment can decrease the temperature range of martensitic transformation and also greatly enhance the mechanical properties of the Nitinol alloy processed by L-PBF. They concluded that a combination of heating-cooling cycling of 350 and 400°C over the temperature range of martensitic transformation can result in a 10 to 15°C decrease in martensitic transformation temperature and also can add another step to the transformation sequence of the structure. They also observed a 7 increase in the maximum stress and dislocation yield stress, as well as a 10 increase in the difference between the dislocation and transformation yield stresses of the developed structure, by applying 10-cycle heating-cooling.”

[176] M. Tsaturyants, V. Sheremetyev, S. Dubinskiy, V. Koma-rov, K. Polyakova, A. Korotitskiy, S. Prokoshkin, E. Borisov, K. Starikov, D. Kaledina, A. Popo-vich, V. Brailovski; Structure and properties of Ti–50.2Ni alloy processed by laser powder bed fusion and subjected to a combination of thermal cycling and heat treatments, Shap. Mem. Superelasticity (2022) 8:16–32.

Remark 1.5. 3.1. Biodegradable metals: Please expand the section. write details about the promising Fe-Mn-Si alloys

The following description about Fe-Mn-Si alloys has been added to section 3.1:

“Fe-Mn-Si alloys, such as the alloy with about 30% mass Mn and 6% mass Si [150], were found to exhibit the shape-memory effect, which looks quite promising for medical and other industrial applications. The martensitic transformation also enhances the mechanical properties of the alloys such as hardness, strength, and fatigue resistance [151].”

[150] Sato A, Kubo H, Tadakatsu M. Mechanical properties of Fe–Mn–Si based SMA and the application. Materials Transactions. 2006;47:571-9.

[151] Sawaguchi T, Maruyama T, Otsuka H, Kushibe A, Inoue Y, Tsuzaki K. Design concept and applications of Fe-Mn-Si-based alloys - from shape-memory to seismic response control. Materials Transactions. 2016;57(3):283-93.

Remark 1.6. The conclusion is too general, it needs to be rewritten

We thank the reviewer for their comments. Concrete conclusions have been added to the Conclusion section:

“…All the AM techniques in the ASTM classification, especially Material Extrusion (e.g., FDM), Directed Energy Deposition (DED), Material Jetting (e.g., Polyjet), PBF (e.g., SLS, SLM, DMLS, and EBM), and Binder Jetting, are not equally developed and used for medical devices and biomaterial fabrication. The capabilities, limitations, pros, and cons of each technique and associated materials (e.g., metals and their alloys, polymers, and ceramics) as well as the considerations for the AM fabrication of biomaterials, such as printing speed, part sizes, degree of anisotropy, achievable resolution, the possibility of embedding cells in feedstock materials, need for support, need for post-processing, and printing costs, are all important factors that need to be taken into account. The success of each of these 3D printing processes relies, to a large extent, on the employment of optimised or suitable process parameters within the capabilities of the available AM machines.

Aside from selecting the proper AM techniques and suitable printing parameters, the microarchitecture design of biomaterials is one of the critical aspects of their development. It is often necessary to design porous or lattice structures for biomedical applications. This implies that pores of a certain morphologies and sizes inside the biomaterials must be fully open and interconnected to allow for the transportation of nutrition and oxygen to the cells.”

Remark 1.7. The article contains a lot of drawings from other published works - this is good, but you need to find out if it is possible to reproduce them from the copyright holder

Permissions for reprinting those figures have been received and stated in the corresponding captions. Figures 2b and 4a have been replaced by figures obtained from other open access research articles. The copyright permissions for all other figures have been provided to the journal.

Reviewer 2 Report

The manuscript under the title: “Additive manufacturing of biomaterials – design principles 2 and their implementation” is in line with the “Materials” journal. The topic undertaken is up-to-date and important. Overall, the review is comprehensive and well-prepared, but it requires some revision. The following elements required to be improved:

·        Abstract: Please add the main findings.

·        Introduction - last paragraph of the Introduction part should explain the main aim of the article and stress the novelty aspect given by review; why this review is needed?

·        Introduction or additional part - lack of information on methodology and research methods. In the review article short information about used keywords, databased or general methodology for the literature review should be presented.

·        Chapter 2. It is a very valuable chapter. Please add there information that the presented methods can be combined.

·        Chapter 2. What about the environmental friendliness and effectiveness of AM?

·        Chapter 3. ‘Among various biocompatible metals and alloys, Ti and its alloys (e.g., Ti6Al4V) are the most popular materials that have been extensively used in biomedical applications’ (lines 331-2), the sources of information are missing; most economic reports show that stainless steel is more popular; please verify and give relevant references.

·        Chapter 3. (lines 410-413), please add more information about potential problems, for example: https://doi.org/10.3390/ma15124213

·        Chapter 4. Add on the end information about other popular polymers in biomedical application such as PMMA and PFTE. What kind of limitation makes AM technologies are not used for the manufacturing of the element from this materials? (please explain in the text).

·        Chapter 5. (lines 985-6), please remove – not necessary for this review.

·        Chapter 5. The content have to be rearrangement. In chapters 3 and 4 the materials are presented according to the type in this chapter according to the production methods. In the one review this element should be coherent. The content should be presented in sub-chapter divided according to the materials types. (IMPOSTANT!).

·        Chapter 5., the reference should be directly behind the author’s name; verify all text, including lines: 1178 – 1214.

·        Please add the chapter information on the 3D printing of composites.

·        Chapter 6. Please try to avoid references in conclusion part; please consider dividing this part to future research / discussion and short summarizing as the most important findings in conclusion part.

·        Conclusions – please add ore connections with the findings from the text

·        Conclusions - This part should not include the references.

·        All text: requires editing, including font size, etc.

·        Add information required by the journal, such as the authors’ contribution, COI, etc.

Author Response

Reviewer #2:

The manuscript under the title: “Additive manufacturing of biomaterials – design principles 2 and their implementation” is in line with the “Materials” journal. The topic undertaken is up-to-date and important. Overall, the review is comprehensive and well-prepared, but it requires some revision. The following elements required to be improved:

Remark 2.1. Abstract: Please add the main findings.

As the review article is focused on the design principals of biomaterials, a critically important finding about the design strategies has been added to Abstract:

“….The design strategies could be categorised as: library-based design, topology optimisation, bio-inspired design, and meta-biomaterials…..”

Remark 2.2. Introduction - last paragraph of the Introduction part should explain the main aim of the article and stress the novelty aspect given by review; why this review is needed?

The main objective has been added at the end of the introduction section:

“The main objective of this review article is to present a clear picture of how this technology can be applied for producing bio-materials of novel designs, what are the challenges and limitations and where the technology is heading… It is intended to stimulate the further development and widespread application of the technology to turn design ideas into implants and other medical devices as well as those in tissue engineering applications.”

Remark 2.3. Introduction or additional part - lack of information on methodology and research methods. In the review article short information about used keywords, databased or general methodology for the literature review should be presented.

The presented literature study concerned a narrative literature review which differs from a systematic review which indeed requires the presentation of the methodology, including search terms, databases used, and inclusion and exclusion criteria. Since we have decided to stick to this narrative tone for our review article, we have decided not to include corresponding general methodology in preparing our review article.

Remark 2.4. Chapter 2. It is a very valuable chapter. Please add there information that the presented methods can be combined.

The following sentence has been added to Chapter 2:

“…In some cases, we may combine two or more of these design methods to obtain a more desirable lattice structure.”

Remark 2.5. Chapter 2. What about the environmental friendliness and effectiveness of AM?

The following sentence has been added to the first paragraph of Introduction:

“AM, being different form other manufacturing methods, such as subtractive and formative methods, results in less scrap and waste of materials and allows for light-weight complex structures, often hollow or porous, thus requiring less material input and energy input during their fabrication and service.”

Remark 2.6. Chapter 3. ‘Among various biocompatible metals and alloys, Ti and its alloys (e.g., Ti6Al4V) are the most popular materials that have been extensively used in biomedical applications’ (lines 331-2), the sources of information are missing; most economic reports show that stainless steel is more popular; please verify and give relevant references.

The sentence has been changed into the following sentence and a reference added:

“Among various biocompatible metals and alloys, Ti and its alloys (e.g., Ti6Al4V) are probably the most extensively studied materials [25].”

Remark 2.7. Chapter 3. (lines 410-413), please add more information about potential problems, for example: https://doi.org/10.3390/ma15124213

We thank the reviewer for the suggestion. We have added the other potential problems related to the AM of magnesium alloys from the suggested reference and other relevant references:

“…which have high flammability, strong chemical activity, low melting point and low evaporation temperature. For some Mg alloys, there is the potential of developing crystallisation cracks because of fusible eutectics, great deformation and stresses due to a high linear thermal expansion coefficient and a broad range of crystallisation temperatures [160].”

[160] Shalomeev V, Tabunshchyk G, Greshta V, Korniejenko K, Duarte Guigou M, Parzych S. Casting welding from magnesium alloy using filler materials that contain scandium. Materials (Basel). 2022;15(12).

Remark 2.8. Chapter 4. Add on the end information about other popular polymers in biomedical application such as PMMA and PFTE. What kind of limitation makes AM technologies are not used for the manufacturing of the element from this materials? (please explain in the text).

We thank the reviewer for the suggestions, and we have added the information about PMMA to subsection 4.3. We have briefly explained why secondary bonding is important in AM and used PMMA as an example. However, we feel that PFTE may not be relevant here, considering its thermal instability and toxicity and it will therefore be beyond the scope of this review. The following paragraph has been added to subsection 4.3:

“Although PMMA (i.e., polymethyl methacrylate) has many favourable characteristics for use in medical applications, such as medicine, denture bases, filling of bone and skull defects, bone implant fixation screws, and vertebrae stabilization, it is not widely employed in 3D printing due to the poor bonding between 3D-printed PMMA and build plate as well as metals [253]. It requires a higher temperature, is susceptible to warp and distortion, needs glue to adhere to the bed, and requires a bed temperature of at least 60 °C [254]. In a study on 3D printed PMMA, infiltration with epoxy was applied to increase the tensile strength and elastic modulus of the printed part from 2.91MPa and 223 MPa to 26.6 MPa and 1190 MPa, respectively [255]. In addition, infiltration with wax was shown to improve the surface quality of the part [255].”

[253] Dimitrova M, Corsalini M, Kazakova R, Vlahova A, Chuchulska B, Barile G, et al. Comparison between conventional PMMA and 3D printed resins for denture bases: A narrative review. Journal of Composites Science. 2022;6(3).

[254] Frazer RQ, Byron Rt Fau - Osborne PB, Osborne Pb Fau - West KP, West KP. PMMA: an essential material in medicine and dentistry. (1050-6934 (Print)).

[255] Polzin C, Spath S, Seitz H. Characterization and evaluation of a PMMA‐based 3D printing process. Rapid Prototyping Journal. 2013;19(1):37-43.

Remark 2.9. Chapter 5. (lines 985-6), please remove – not necessary for this review.

The reviewer’s suggestion has been followed and the indicated lines have been removed.

Remark 2.10. Chapter 5. The content have to be rearrangement. In chapters 3 and 4 the materials are presented according to the type in this chapter according to the production methods. In the one review this element should be coherent. The content should be presented in sub-chapter divided according to the materials types. (IMPOSTANT!).

We thank the reviewer for the suggestion and agree with the statement; however, changing the layout of the chapter on ceramics is not possible in this review. We have decided to preserve the chapter’s format, since there are vastly different types of materials for metals and polymers and applicable 3D printing technology may not be the same for different material types. In the case of ceramics, however, the variety of materials is not that large, and the AM technology is mostly applicable to various ceramics in the same manner.

Remark 2.11. Chapter 5., the reference should be directly behind the author’s name; verify all text, including lines: 1178 – 1214.

The reviewer’s suggestion has been followed and corrections have been made.

Remark 2.12. Please add the chapter information on the 3D printing of composites.

The reviewer’s suggestion has been followed and the following section has been added to the chapter 4:

“4.4. Composites

Polymer composites or polymer matrix composites are obtained by incorporating reinforcements of particles, fibres or nanomaterials into polymers. This results in better mechanical properties and functionality. Such composites are extensively used in a wide range of medical applications including dental treatments, regenerative medicine, and tissue engineering. The materials that are used for these applications must be biocompatible and have the required mechanical and physical properties. A bio-composite is also classified as a composite which contains natural reinforcing fibres [420]. AM of composite structures has attracted a lot of attention recently due to having flexibility and the ability to produce high-performance products while being able to control the geometry of composite structures and constituents and minimising the waste [421].

4.4.1. Particle-reinforced polymer composites

Particles can be easily and economically incorporated into the polymer matrix either in the powder form or liquid form, depending on the method of 3D printing. They can greatly enhance the physical and mechanical properties of the product; for example adding iron or copper [422] particles or glass beads can improve the tensile modulus of the polymer matrix [423].

4.4.2. Fibre-reinforced polymer composites

Glass fibres [424] and carbon fibres [425, 426] are the most favourite reinforcements used for polymer matrix composites to enhance their mechanical properties. In addition to the type of reinforcement, the orientation and void fraction of the fibres determine the properties of the final printed product [427]. During the 3D printing process, some voids may be formed, which can affect the mechanical properties of the final 3D printed structure [428]. The porosity in the 3D printed parts due to voids can be significantly reduced by adding expandable microspheres to the polymer [429]. Up to now, it has been nearly impossible to print continuous fibres and only short fibres could be 3D printed. Recently, there have been major developments in establishing the relationship between process parameters and printed composite specimens [430, 431].

Another major development concerns shape-memory polymer composites that can expand the 3D printing to 4D printing technology by incorporating the time factor. These composite are of great interest due to the ability to recover deformation [432]. In addition to temperature-responsive shape-memory composites, water-responsive shape-memory composites have also been developed by applying AM methods [433]. In the cited study, a composite composed of cellulose fibrils and acrylamide, changed its shape when immersed in water.

4.4.3. Nanocomposites

Nanomaterials have also been incorporated into matrix materials to enhance their mechanical properties. Shofner et al. [434] demonstrated that an addition of 10 wt% carbon nanofibre could increase the tensile strength of 3D printed parts by up to 39%, although the elongation reduced and brittleness increased. In another study, an addition of just 0.2% graphene oxide in an SLA fabricated photopolymer composite caused increases of 62% in tensile strength and 12.8% in elongation [435], which is quite remarkable. Introduction of nanomaterials, such as carbon nanotubes, can also significantly improve the electrical properties in addition to the mechanical properties of the composite [436].

Functionally graded polymer nanocomposites have been realised by 3D printing. This can be performed by introducing different volume fractions of nanomaterials to specific locations of the structure of the part [437].”

[420] Zagho MM, Hussein EA, Elzatahry AA. Recent overviews in functional polymer composites for biomedical applications. Polymers. 2018;10(7):739.

[421] Wang X, Jiang M, Zhou Z, Gou J, Hui D. 3D printing of polymer matrix composites: A review and prospective. Composites Part B: Engineering. 2017;110:442-58.

[422] Nikzad M, Masood SH, Sbarski I. Thermo-mechanical properties of a highly filled polymeric composites for Fused Deposition Modeling. Materials & Design. 2011;32(6):3448-56.

[423] Chung H, Das S. Processing and properties of glass bead particulate-filled functionally graded Nylon-11 composites produced by selective laser sintering. Materials Science and Engineering: A. 2006;437(2):226-34.

[424] Zhong W, Li F, Zhang Z, Song L, Li Z. Short fiber reinforced composites for fused deposition modeling. Materials Science and Engineering: A. 2001;301(2):125-30.

[425] Ning F, Cong W, Qiu J, Wei J, Wang S. Additive manufacturing of carbon fiber reinforced thermoplastic composites using fused deposition modeling. Composites Part B: Engineering. 2015;80:369-78.

[426] Griffini G, Invernizzi M, Levi M, Natale G, Postiglione G, Turri S. 3D-printable CFR polymer composites with dual-cure sequential IPNs. Polymer. 2016;91:174-9.

[427] Tekinalp HL, Kunc V, Velez-Garcia GM, Duty CE, Love LJ, Naskar AK, et al. Highly oriented carbon fiber–polymer composites via additive manufacturing. Composites Science and Technology. 2014;105:144-50.

[428] Le Duigou A, Castro M, Bevan R, Martin N. 3D printing of wood fibre biocomposites: From mechanical to actuation functionality. Materials & Design. 2016;96:106-14.

[429] Wang J, Xie H, Weng Z, Senthil T, Wu L. A novel approach to improve mechanical properties of parts fabricated by fused deposition modeling. Materials & Design. 2016;105:152-9.

[430] Van Der Klift F, Koga Y, Todoroki A, Ueda M, Hirano Y, Matsuzaki R. 3D printing of continuous carbon fibre reinforced thermo-plastic (CFRTP) tensile test specimens. Open Journal of Composite Materials. 2016;6(01):18.

[431] Tian X, Liu T, Yang C, Wang Q, Li D. Interface and performance of 3D printed continuous carbon fiber reinforced PLA composites. Composites Part A: Applied Science and Manufacturing. 2016;88:198-205.

[432] Lu H, Yao Y, Huang WM, Leng J, Hui D. Significantly improving infrared light-induced shape recovery behavior of shape memory polymeric nanocomposite via a synergistic effect of carbon nanotube and boron nitride. Composites Part B: Engineering. 2014;62:256-61.

[433] Sydney Gladman A, Matsumoto EA, Nuzzo RG, Mahadevan L, Lewis JA. Biomimetic 4D printing. Nature materials. 2016;15(4):413-8.

[434] Shofner M, Lozano K, Rodríguez‐Macías F, Barrera E. Nanofiber‐reinforced polymers prepared by fused deposition modeling. Journal of applied polymer science. 2003;89(11):3081-90.

[435] Lin D, Jin S, Zhang F, Wang C, Wang Y, Zhou C, et al. 3D stereolithography printing of graphene oxide reinforced complex architectures. Nanotechnology. 2015;26(43):434003.

[436] Guo S-z, Yang X, Heuzey M-C, Therriault D. 3D printing of a multifunctional nanocomposite helical liquid sensor. Nanoscale. 2015;7(15):6451-6.

[437] Chung H, Das S. Functionally graded Nylon-11/silica nanocomposites produced by selective laser sintering. Materials Science and Engineering: A. 2008;487(1-2):251-7.

Remark 2.13. Chapter 6. Please try to avoid references in conclusion part; please consider dividing this part to future research / discussion and short summarizing as the most important findings in conclusion part.

The references have been removed from the Conclusion section. Future research directions have been highlighted in the last paragraph of the manuscript.

Remark 2.14. Conclusions – please add ore connections with the findings from the text

The following paragraphs have been added to the Conclusion section:

“…All the AM techniques in the ASTM classification, especially Material Extrusion (e.g., FDM), Directed Energy Deposition (DED), Material Jetting (e.g., Polyjet), PBF (e.g., SLS, SLM, DMLS, and EBM), and Binder Jetting, are not equally developed and used for medical devices and biomaterial fabrication. The capabilities, limitations, pros, and cons of each technique and associated materials (e.g., metals and their alloys, polymers, and ceramics) as well as the considerations taken into the AM fabrication of biomaterials such as printing speed, part sizes, degree of anisotropy, achievable resolution, the possibility of embedding cells in feedstock materials, need for support, need for post-processing, and printing costs, all are important factors that need to be taken into account. The success of each of these 3D printing processes relies, to a large extent, on the employment of optimised or suitable process parameters within the capabilities of the available AM machines.

Aside from selecting the proper AM techniques and suitable printing parameters, the microarchitecture design of biomaterials is one of the critical aspects of their development. It is often necessary to design porous or lattice structures for biomedical applications. This implies that pores of certain morphologies and sizes insider the biomaterials must be fully open and interconnected to allow for the transportation of nutrition and oxygen to the cells…”

Remark 2.15. Conclusions - This part should not include the references.

The references have been removed from the Conclusion section.

Remark 2.16. All text: requires editing, including font size, etc.

All co-authors have carefully checked the text and edited the text.

Remark 2.17. Add information required by the journal, such as the authors’ contribution, COI, etc.

Sections “Competing interests” and “Authors’ contribution” have been added to the revised manuscript.

Reviewer 3 Report

The paper discussion interesting note on design topics, but needs significant improvements. Please follow the comments carefully.

1.      Add some quantitative results to the abstract.

2.      What are the future works for this research. Please clarify it in the separated section.

3.      Add more discussion about figure 3.

4.      Provide a thorough discussion of reported results and discussions.

5.      Add more results to your conclusion.

6.      Laser absorptivity in AM is important in design which shows the quality of the parts. Please read and add the following ref in this area. “The effect of absorption ratio on meltpool features in laser-based powder bed fusion of IN718”.

7.      The introduction needs to be updated with the following new references.

·        Topological design of the hybrid structure with high damping and strength efficiency for additive manufacturing

·        A computational approach from design to degradation of additively manufactured scaffold for bone tissue engineering application

·        Design, additive manufacturing and component testing of pneumatic rotary vane actuators for lightweight robots

Author Response

Reviewer #3:

The paper discussion interesting note on design topics, but needs significant improvements. Please follow the comments carefully.

Remark 3.1. Add some quantitative results to the abstract.

The review covers a wide range of biomaterials and its primary focus is on design principles for AM. We have accordingly added the following line to Abstract to describe the most used design strategies.

“… The design strategies could be categorised as: library-based design, topology optimisation, bio-inspired design, and meta-biomaterials…..”

Remark 3.2. What are the future works for this research. Please clarify it in the separated section.

We have highlighted the future research directions in the last paragraph of the revised manuscript and clarified it with:

“… and require more research in the future.”

Remark 3.3. Add more discussion about figure 3.

Following lines have been added to the last paragraph of section 4.3.2. Discussion about figures 3f-I can be found in section 4.3.5.

“…When heated above their glass transition temperature, extruded PLA filaments (i.e., 3D printed) shorten in the printing direction and thicken simultaneously. By rationally placing printed filaments into a multi-layer construct, a complex 3D structure can be obtained after the flat construct is thermally activated [175]… flat constructs to pre-programmed 3D shapes… which employ design strategies such as instability-driven pop-up (Figure 3a), self-folding origami (Figure 3b), and sequential shape-shifting (Figure 3c).”

Remark 3.4. Provide a thorough discussion of reported results and discussions.

We attempted to include more discussion and information in various sections of the manuscript. They are highlighted in the revised manuscript.

Remark 3.5. Add more results to your conclusion.

The following sentences have been added to the Conclusion section:

“…All the AM techniques in the ASTM classification, especially Material Extrusion (e.g., FDM), Directed Energy Deposition (DED), Material Jetting (e.g., Polyjet), PBF (e.g., SLS, SLM, DMLS, and EBM), and Binder Jetting, are not equally developed and used for medical devices and biomaterial fabrication. The capabilities, limitations, pros, and cons of each technique and associated materials (e.g., metals and their alloys, polymers, and ceramics) as well as the considerations taken into the AM fabrication of biomaterials such as printing speed, part sizes, degree of anisotropy, achievable resolution, the possibility of embed-ding cells in feedstock materials, need for support, need for post-processing, and printing costs, all are important factors that need to be taken into account. The success of each of these 3D printing processes relies, to a large extent, on the employment of optimised or suitable process parameters within the capabilities of the available AM machines.

Aside from selecting the proper AM techniques and suitable printing parameters, the microarchitecture design of biomaterials is one of the critical aspects of their development. It is often necessary to design porous or lattice structures for biomedical applications. This implies that pores of certain morphologies and sizes inside the biomaterials must be fully open and interconnected to allow for the transportation of nutrition and oxygen to the cells…”

Remark 3.6. Laser absorptivity in AM is important in design which shows the quality of the parts. Please read and add the following ref in this area. “The effect of absorption ratio on meltpool features in laser-based powder bed fusion of IN718”.

We thank the reviewer for the suggestion. The reference is mentioned in the text as:

[179] M. Khorasani, A. Ghasemi, M. Leary, E. Sharabian, L. Cordova, I. Gibson, D. Downing, S. Bateman, M. Brandt, B. Rolfe, The effect of absorption ratio on meltpool features in laser-based powder bed fusion of IN718, Optics & Laser Technology 153 (2022) 108263.

Remark 3.7. The introduction needs to be updated with the following new references.

-           Topological design of the hybrid structure with high damping and strength efficiency for additive manufacturing

-           A computational approach from design to degradation of additively manufactured scaffold for bone tissue engineering application

-           Design, additive manufacturing and component testing of pneumatic rotary vane actuators for lightweight robots

Reviewer’s suggestions have been followed and the following references have been added to the reference list:

[3] G. Dämmer, H. Bauer, R. Neumann, Z. Major, Design, additive manufacturing and component testing of pneumatic rotary vane actuators for lightweight robots, Rapid Prototyping Journal 28(11) (2022) 20-32.

[23] M. Kumar, S.S. Mohol, V. Sharma, A computational approach from design to degradation of additively manufactured scaffold for bone tissue engineering application, Rapid Prototyping Journal  (2022) (ahead-of-print).

[65] R. Wang, Y. Chen, X. Peng, N. Cong, D. Fang, X. Liang, J. Shang, Topological design of the hybrid structure with high damping and strength efficiency for additive manufacturing, Rapid Prototyping Journal  (2022) (ahead-of-print).

Round 2

Reviewer 1 Report

A revised version can be published in present form

Reviewer 3 Report

The paper is ready to publish.